# SUV39 SET domains mediate crosstalk of heterochromatic histone marks

Alessandro Stirpe[1†], Nora Guidotti[2‡], Sarah J Northall[3,4], Sinan Kilic[2§], Alexandre Hainard[5], Oscar Vadas[6#], Beat Fierz[2], Thomas Schalch[1,3,4*]

[1]Department of Molecular Biology, Faculty of Science, University of Geneva, Geneva, Switzerland; [2]Institute of Chemical Sciences and Engineering (ISIC), Ecole Polytechnique Fédérale de Lausanne (EPFL), Lausanne, Switzerland; [3]Leicester Institute for Structural and Chemical Biology, University of Leicester, Leicester, United Kingdom; [4]Department of Molecular and Cell Biology, University of Leicester, Leicester, United Kingdom; [5]University Medical Center, University of Geneva, Geneva, Switzerland; [6]School of Pharmaceutical Sciences, Faculty of Science, University of Geneva, Geneva, Switzerland

*For correspondence:
thomas.schalch@le.ac.uk

Present address: †Wellcome Trust Centre for Cell Biology, Institute of Cell Biology, School of Biological Sciences, University of Edinburgh, Edinburgh, United Kingdom; ‡Athebio AG, Zurich, Switzerland; §Novo Nordisk Foundation Center for Protein Research, University of Copenhagen, Copenhagen, Denmark; #Protein Platform, Faculty of Medicine, University of Geneva, Geneva, Switzerland

Competing interest: The authors declare that no competing interests exist.

**Abstract** The SUV39 class of methyltransferase enzymes deposits histone H3 lysine 9 di- and trimethylation (H3K9me2/3), the hallmark of constitutive heterochromatin. How these enzymes are regulated to mark specific genomic regions as heterochromatic is poorly understood. Clr4 is the sole H3K9me2/3 methyltransferase in the fission yeast *Schizosaccharomyces pombe,* and recent evidence suggests that ubiquitination of lysine 14 on histone H3 (H3K14ub) plays a key role in H3K9 methylation. However, the molecular mechanism of this regulation and its role in heterochromatin formation remain to be determined. Our structure-function approach shows that the H3K14ub substrate binds specifically and tightly to the catalytic domain of Clr4, and thereby stimulates the enzyme by over 250-fold. Mutations that disrupt this mechanism lead to a loss of H3K9me2/3 and abolish heterochromatin silencing similar to *clr4* deletion. Comparison with mammalian SET domain proteins suggests that the Clr4 SET domain harbors a conserved sensor for H3K14ub, which mediates licensing of heterochromatin formation.

## Introduction

Histone methylation plays a critical role in regulating and organizing genomes to maintain genome integrity and establish appropriate gene regulation programs (*Greer and Shi, 2012*; *Janssen et al., 2018*). Histone H3 lysine 9 di- and trimethylation (H3K9me2/3) are highly conserved hallmarks of heterochromatin, where they provide a platform for the recruitment of chromatin effector complexes. H3K9me2/3 marks are dysregulated in various cancers, in neurological diseases and viral latency, and are targeted for therapeutic approaches (*Kaniskan et al., 2018*).

The SUV39 clade of the SET-domain family of protein lysine methyltransferases represents the enzyme class depositing H3K9me2/3 (*Dillon et al., 2005*). They share a SET domain as a catalytic core, which builds on a series of curved β-sheets that are sandwiched between distinct pre- and post-SET domains, providing structural elements based on zinc-finger motifs. The SUV39 enzymatic activity facilitates the transfer of a methyl group from S-adenosyl-L-methionine (SAM) to the $\varepsilon$-amino group of lysine 9 on histone H3 with high specificity, thereby converting the cofactor to S-adenosyl-L-homocysteine (SAH).

The mechanisms regulating SUV39 proteins to specifically deposit H3K9me2/3 in heterochromatic regions are not well understood. The fission yeast *Schizosaccharomyces pombe* heterochromatin system, closely related to the metazoan systems, with its sole SUV39 protein Clr4, serves as a

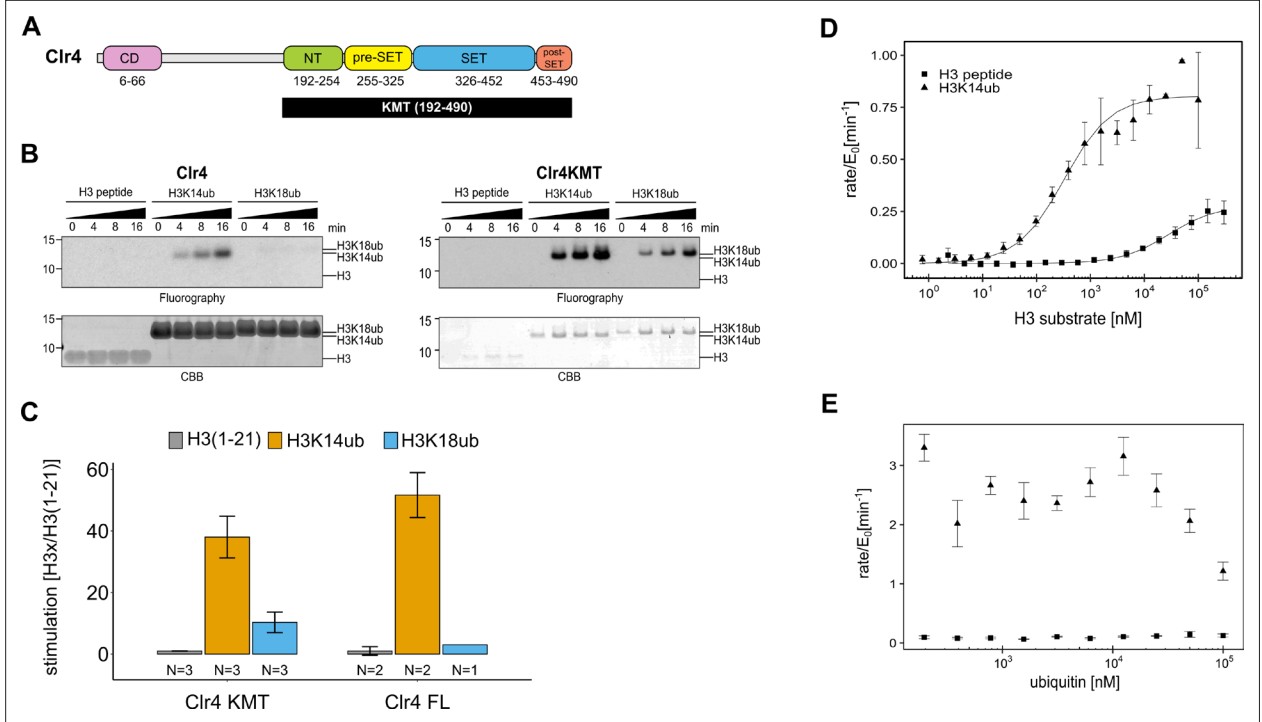

**Figure 1.** The catalytic domain of Clr4 senses the H3K14ub mark. (**A**) Domain organization of the Clr4 protein. CD: chromodomain; NT: N-terminal domain; SET: Su-var39/enhancer of zeste/trithorax domain; KMT: lysine methyltransferase domain. (**B**) Methyltransferase time course on H3K14ub versus unmodified H3 peptides shows strong stimulation of full-length Clr4 and the isolated KMT domain by H3K14ub. Peptide concentration was held constant at 20 μM, $^3$H-SAM was used as methyl donor with 20 nM enzyme. CBB: Coomassie Brilliant Blue. (**C**) Stimulation of methyltransferase activity of Clr4 (20 nM) comparing indicated substrates with unmodified H3 peptide by radiometric filter binding assay with $^3$H-SAM as methyl donor. (**D**) Michaelis–Menten kinetics of SAH production of the Clr4 KMT domain on unmodified versus H3K14ub peptides. Measured using TR-FRET competition assay (Cisbio EPIgeneous Methyltransferase Assay kit). (**E**) Ubiquitin competition assay demonstrates the specificity of Clr4 for ubiquitin and shows that covalent linkage in cis is required for activation. (**C–E**) Error bars indicate standard error of the mean, N = 3 unless indicated.

The online version of this article includes the following figure supplement(s) for figure 1:

**Figure supplement 1.** H3K14ub activates Clr4 enzymatic activity.

paradigm for studying the role of this enzyme class in heterochromatin establishment and mainte-nance. Ectopic tethering experiments with Clr4 have established that H3K9me2/3 can support stable epigenetic transmission of transcriptional states (*Audergon et al., 2015*; *Ragunathan et al., 2015*). Clr4 has further served as a valuable tool to understand its animal orthologs, which share a very similar domain architecture consisting of an N-terminal chromodomain and a C-terminal catalytic SET domain (*Figure 1A*). The chromodomain binds the H3K9me2/3 mark and is critical for spreading the H3K9me2/3 mark across genomic domains (*Melcher et al., 2000*; *Müller et al., 2016*; *Zhang et al., 2008*). Recent evidence also suggests that an autoinhibitory loop in the post-SET domain regulates spreading of H3K9me2/3 by Clr4 (*Iglesias et al., 2018*).

Clr4 is a key part of *S. pombe* heterochromatin, which is found at pericentromeres, telomeres, and the mating type locus and depends on the tightly interconnected pathways of the nuclear RNA inter-ference machinery and an array of chromatin modifiers (*Grewal, 2010*). Small RNAs are generated from a limited set of repeat sequences in heterochromatic regions and guide the RNAi machinery in the form of the RNA-induced initiation of transcriptional gene silencing complex (RITS) to nascent heterochromatic transcripts (*Verdel et al., 2004*; *Bühler et al., 2007*; *Schalch et al., 2011*; *Shimada et al., 2016*). RITS recruits Clr4 and its associated Cullin4-RING ubiquitin ligase complex, the CLRC complex (*Zhang et al., 2008*; *Bayne et al., 2010*), thereby driving the deposition of H3K9me2/3 marks, which provides a platform for binding of HP1 proteins that in turn recruit further transcriptional gene silencing complexes (*Motamedi et al., 2008*; *Fischer et al., 2009*; *Leopold et al., 2019*). Heter-ochromatin formation is further strongly dependent on deacetylation of H3K14 (*Alper et al., 2013*;

*Bjerling et al., 2002*; *Buscaino et al., 2013*; *Yamada et al., 2005*), with H3K14 acetylation being strongly associated with active gene expression in euchromatin (*Wang et al., 2012*).

Even though it has long been recognized that *rik1* plays an essential role in the *S. pombe* heterochromatin system (*Ekwall and Ruusala, 1994*; *Allshire et al., 1995*; *Nakayama et al., 2001*; *Zhang et al., 2008*), its molecular mechanism has remained elusive. Rik1 is a paralog of the DNA repair protein DDB1 and forms the central scaffold of the CLRC complex that includes the cullin protein Cul4, the WD-40 β-propeller proteins Dos1/Raf1, the RING finger protein Pip1, as well as the replication focus targeting sequence (RFTS) protein Dos2/Raf2 and Clr4 (*Hong et al., 2005*; *Horn et al., 2005*; *Jia et al., 2005*; *Li et al., 2005*; *Thon et al., 2005*). CLRC features the hallmarks of an intact CRL4-type E3 ligase, which uses the subunit Dos1/Raf1 as a substrate adapter (*Buscaino et al., 2012*; *Horn et al., 2005*; *Jia et al., 2005*; *Kuscu et al., 2014*). Recent work has revealed that the preferred substrate for ubiquitylation by CLRC is lysine 14 on histone H3, yielding H3K14ub (*Oya et al., 2019*), and that H3K14ub controls the activity of Clr4. However, how H3K14ub regulates Clr4 and its role in heterochromatin formation remain to be determined.

We combined biochemical and structural methods to decipher the molecular mechanism governing the stimulation of Clr4 by H3K14ub. These experiments identified a ubiquitin binding region (UBR) in the catalytic domain of Clr4, which mediates H3K14ub binding, and mutations that disrupt this interaction interface lose heterochromatin function. This reveals a critical regulatory mechanism that uses the SET domain of Clr4 for control of heterochromatin formation by an epigenetic crosstalk.

## Results

### The catalytic domain of Clr4 senses the presence of H3K14-linked ubiquitin

To understand the enzymatic reaction underlying the stimulation of Clr4 by H3K14ub, we set out to determine the domains of Clr4 that mediate this effect. Clr4's domain architecture comprises an N-terminal chromodomain, which is connected by a linker region to the catalytic lysine methyltransferase (KMT) domain (*Figure 1A*). The KMT domain contains the N-terminal, pre-SET, SET, and post-SET regions and its structure has been determined (*Min et al., 2002*; *Iglesias et al., 2018*). To determine the role of the chromodomain and linker region, we recombinantly expressed both the full-length Clr4 and the isolated KMT domain and quantified their methyltransferase activity using two different assays: (1) incorporation of tritium-labeled methyl groups from a $^3$H-SAM donor into histone proteins and (2) fluorescence-based measurement of SAH generation (Cisbio EPIgeneous Methyltransferase Assay kit). As substrates we used unmodified H3 peptide (1–21) and H3 peptide ubiquitinated on K14 (H3K14ub) or K18 (H3K18ub) generated by native chemical ligation procedures. These branched peptides are identical to a native ubiquitin linkage except for a glycine to alanine change in the C-terminal residue of ubiquitin that is covalently linked to K14 (*Figure 1—figure supplement 1A*).

*Figure 1B* shows fluorographs of methyltransferase assays for Clr4 full-length and the KMT domain, which both manifest a similar degree of strong stimulation by H3K14ub (*Figure 1C*, *Figure 1—figure supplement 1E–H*). To get a sense for the steric requirements involved in stimulation by H3K14ub, we tested H3K18ub, the closest downstream lysine on the H3 tail. H3K18ub is a significantly poorer substrate than H3K14ub, demonstrating that the position of the ubiquitin modification on the H3 tail is critical for optimal Clr4 stimulation and that covalent linkage generating high local ubiquitin concentration is not sufficient. Quantification of the stimulation by the different substrates further showed that H3K14ub elicits a 40–50-fold boost in activity at a substrate concentration of 20 μM (*Figure 1C*). These experiments confirm the observation by *Oya et al., 2019* that the H3K14ub substrate triggers a dramatic and specific increase in the methyltransferase activity of Clr4. However, in contrast to the previous study, we observe that the KMT domain is sufficient to mediate this regulatory mechanism. Since the primary effect of H3K14ub traces to the catalytic domain, we decided to further investigate the stimulation mechanism using the isolated KMT domain.

To fully characterize the H3K14ub-mediated stimulation, we measured enzyme kinetics of Clr4KMT on H3K14ub vs. the unmodified peptide (*Figure 1D*). An approximately hundred-fold difference for the Michaelis–Menten ($K_M$) constants was measured with 0.33 ± 0.06 μM for H3K14ub and 28.1 ± 8.0 μM for H3. The turnover number ($k_{cat}$) was about three times higher in the presence of H3K14ub (0.81 ± 0.03 min$^{-1}$) compared with unmodified H3 (0.28 ± 0.02 min$^{-1}$). This leads to an increase in

**Table 1.** Enzyme kinetics.

| Clr4KMT | Substrate | $K_M$(μM) | $k_{cat}$(min$^{-1}$) | $k_{cat}$/$K_M$(mM$^{-1}$ min$^{-1}$) | Figure |
|---------|-----------|-----------|----------------------|--------------------------------------|--------|
| WT | H3(1–21) | 28.1 ± 8.0 | 0.277 ± 0.024 | 9.86 ± 2.18 | *Figure 1D* |
| WT | H3K14ub | 0.329 ± 0.060 | 0.809 ± 0.029 | 2456 ± 405 | *Figure 1D* |
| WT | H3(1–19) | 124 ± 26 | 0.713 ± 0.085 | 5.75 ± 0.59 | *Figure 3G* |
| WT | H3K14ub | 0.234 ± .0068 | 3.85 ± 0.29 | 16459 ± 3,796 | *Figure 3G* |
| 3FA | H3(1–19) | 76.8 ± 17.5 | 0.406 ± 0.045 | 5.29 ± 0.68 | *Figure 3G* |
| 3FA | H3K14ub | 4.00 ± 1.60 | 2.49 ± 0.32 | 623 ± 180 | *Figure 3G* |
| GS253 | H3(1–19) | 121 ± 29 | 0.548 ± 0.071 | 4.53 ± 0.52 | *Figure 3G* |
| GS253 | H3K14ub | 10.4 ± 1.2 | 3.83 ± 0.21 | 370 ± 25 | *Figure 3G* |

Values represent fitting estimates and corresponding standard error.

overall enzymatic efficiency measured by the specificity constant $k_{sp}$ = $k_{cat}$/$K_M$ of 250-fold (*Table 1*). Comparison of the kinetic parameters between H3K14ub and H3 substrate indicates that the presence of ubiquitin on lysine 14 leads to a tighter enzyme-substrate complex and to conformational changes in the active site that increase the rate of the methyltransferase reaction.

To determine whether H3K14ub uses an allosteric site for ubiquitin on Clr4, we challenged the methyltransferase reaction with increasing amounts of free ubiquitin. While we observed no significant increase in activity for unmodified H3, we observed a drop in the activity for H3K14ub at high concentrations of free ubiquitin (*Figure 1E*). This experiment failed to produce evidence of an allosteric site for free ubiquitin on Clr4, and we conclude that the stimulation of $k_{cat}$ is likely to depend on an induced-fit mechanism triggered by binding of H3K14ub to the Clr4KMT domain.

## H3K14 ubiquitin mark affects the structural dynamics of Clr4

To understand the structural basis for the regulation of Clr4 by H3K14ub, we performed hydrogen/deuterium exchange coupled to mass spectrometry (HDX-MS) analysis on the free Clr4KMT domain and on Clr4KMT in complex with H3- and H3K14ub peptides. HDX-MS measures protein dynamics based on the rate of exchange of protein amide protons with the solvent (*Kochert et al., 2018*). Changes in HDX rates upon complex formation identify regions of the protein that are affected by the formation of the complex. Comparing the dynamics for Clr4KMT alone and in the presence of excess H3K14ub or unmodified H3 peptides, we observed a strong and unique reduction of the HDX rate for residues 243–261 of Clr4 in the presence of H3K14ub (*Figure 2A and B*, *Figure 2—figure supplement 1*, *Supplementary files 2-4*). A further region between residues 291–305 showed significant amide protection by both H3 and H3K14ub substrates, the latter peptide showing a more intense protection (*Figure 2B*, bottom graph). These results suggest that the H3 peptide interacts with residues 291–305, while the ubiquitin moiety binds to a region involving residues 243–261. The H3K14ub binding region identified by HDX-MS maps to the NT domain just before it transitions into the pre-SET domain, and we will refer to this region as UBR (*Figure 2A*). The UBR forms a ridge along the 'back' of Clr4, opposite to the active site pockets where cofactor and the substrate peptide bind (*Figure 2C*).

We also compared HDX-MS rates of isolated H3K14ub peptide with the H3K14ub-Clr4 complex and found that amino acids 25–43 on ubiquitin were protected from exchange upon interaction with Clr4 (*Figure 2D and E*, *Figure 2—figure supplement 1*, *Supplementary file 3*). These residues map to the α-helix of ubiquitin, indicating that this surface interacts with Clr4.

## UBR mutants lose affinity for H3K14ub

To determine the functional importance of the UBR, we designed mutations that specifically disrupt the Clr4-H3K14ub interaction. While substitutions of residues 243–251 yielded unstable protein, substitution of residues 253–256 (sequence DPNF to GGSG, referred to as Clr4-GS253) resulted in a stable protein. Based on the Clr4 structure and sequence conservation, we further chose to mutate

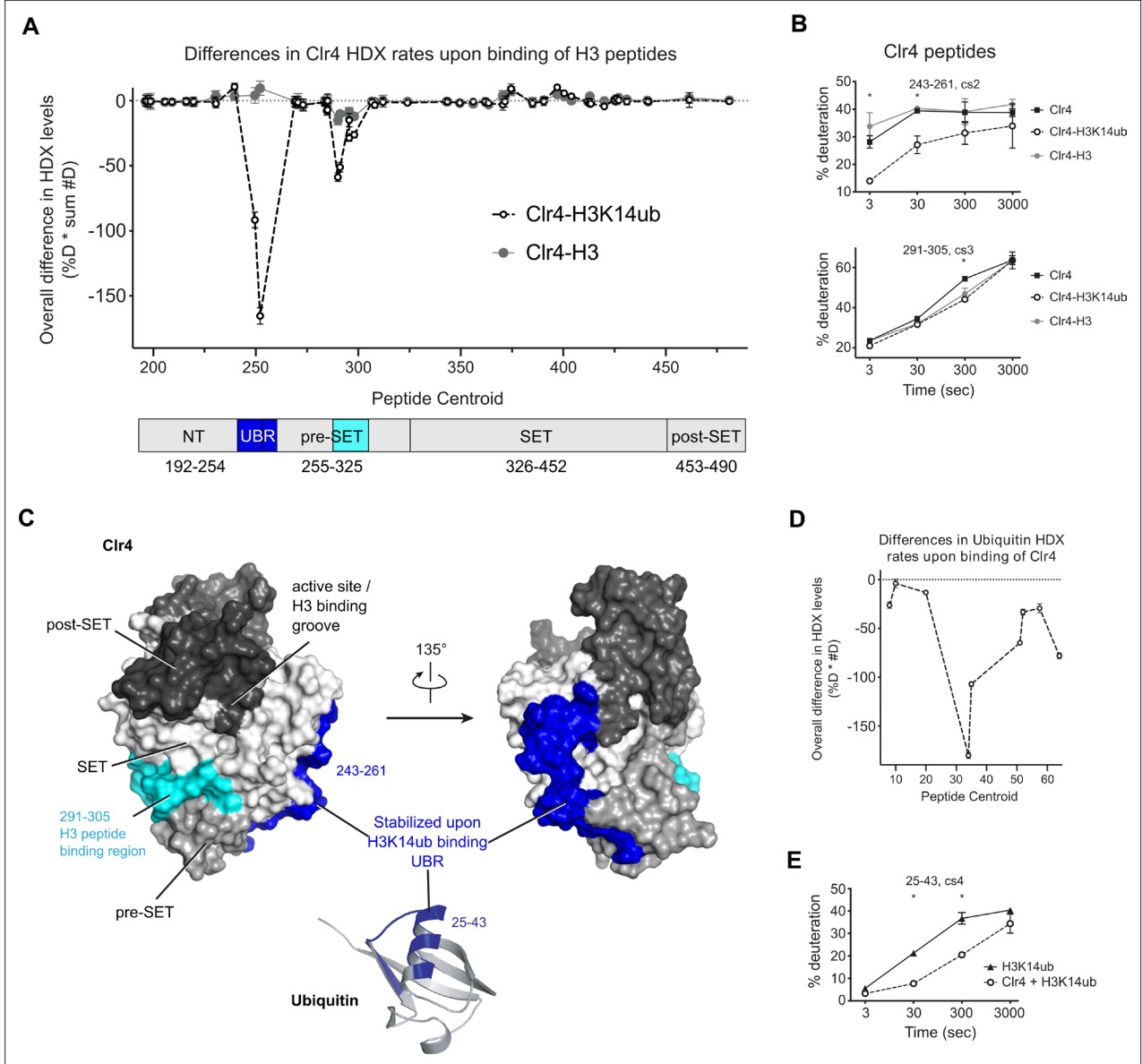

**Figure 2.** H3K14ub binding stabilizes residues 243–262 on Clr4. (**A**) Differences in Clr4 hydrogen/deuterium exchange (HDX) rates upon binding of H3 and H3K14ub peptides are shown for each analyzed peptide. Values are plotted as the product of [deuteration percentage] * [number of deuterons] to minimize the influence of peptide length on the results. Domain diagram indicates the regions showing stabilization upon interaction with H3 peptides in general (cyan) and more specifically with H3K14ub (blue). (**B**) Uptake plot for two peptides representative of regions showing differences in HDX rate of Clr4 upon peptide binding. (**C**) Surface representation of Clr4 structure (PDBID:6BOX) and cartoon representation of ubiquitin (PDBID:1UBQ). UBR: ubiquitin binding region. (**D**) Differences in ubiquitin HDX rates for the H3-K14ub peptide in the absence and presence of Clr4. Results are shown as in (**A**). (**E**) Uptake plot for peptide 25–43 of ubiquitin linked to H3K14 showing a marked reduction in HDX rate at 30 s and 300 s incubation time in deuterated buffer.

The online version of this article includes the following figure supplement(s) for figure 2:

**Figure supplement 1.** H3K14ub binding stabilizes residues 243–262 on Clr4.

three phenylalanines that intersect orthogonally with the GS253 mutations on the surface of Clr4 (F256A, F310A, and F427A, referred to as Clr4-3FA).

We used isothermal titration calorimetry (ITC) to determine the effect of the mutations on the affinity of the Clr4-H3K14ub interaction. Consistent with the low $K_M$ observed previously, Clr4 binds to the H3K14ub peptide with a dissociation constant ($K_d$) of 80 ± 6 nM in a reaction that is dominated by enthalpy (*Figure 3A*). In contrast, Clr4-GS253 and Clr4-3FA showed complete loss of binding to

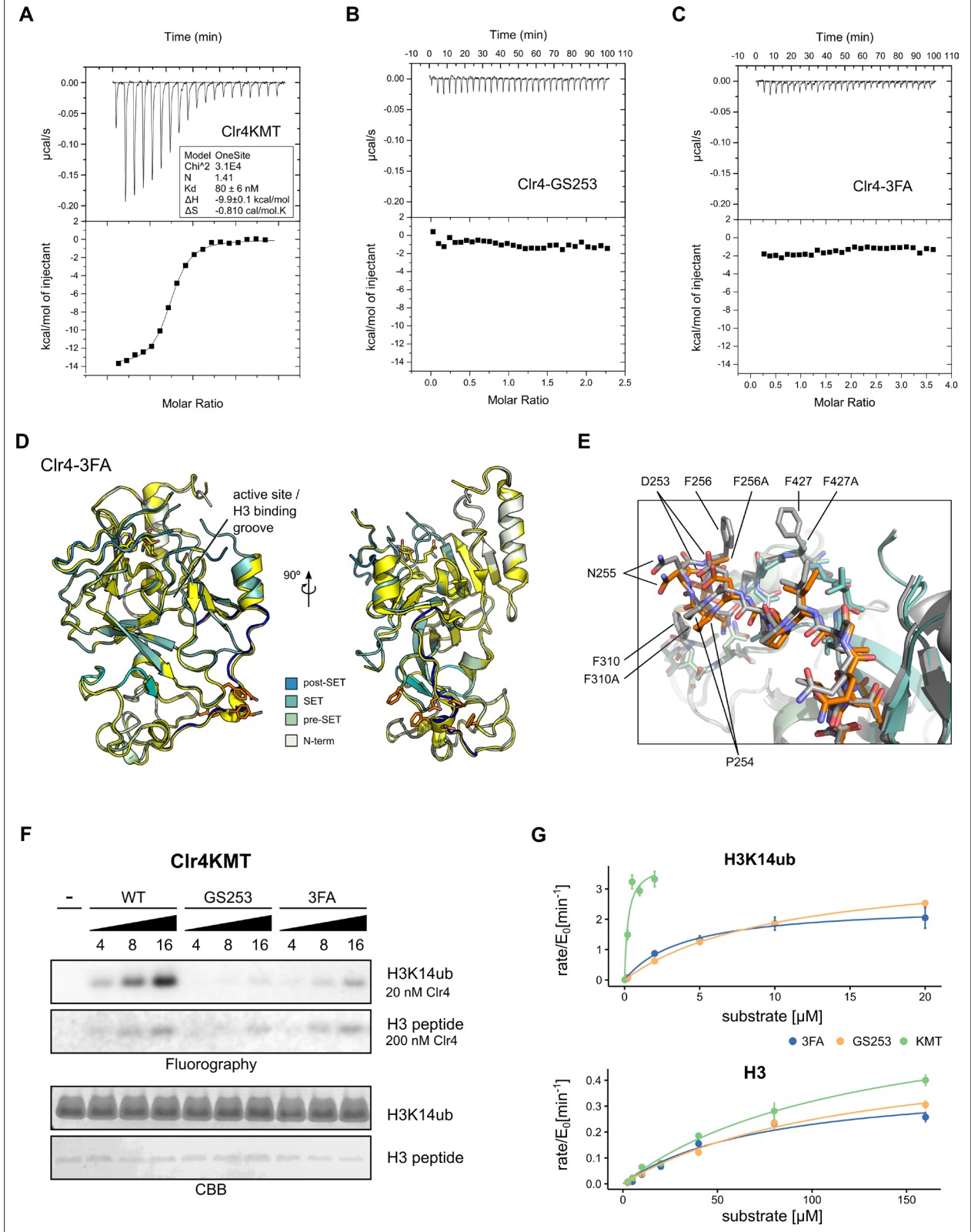

**Figure 3.** Mutants in the Clr4-H3K14ub interface are defective for substrate binding. (**A–C**) Thermodynamic parameters of H3K14ub substrate binding to wild-type and mutant Clr4KMT proteins were measured using isothermal titration calorimetry (ITC). Heat rates (top panel) were integrated and plotted as a binding isotherm (bottom panel). Fit to a one-site model is shown as a solid line where fitting was possible. (**D**) Overview and detail view of superimposition of Clr4-3FA crystal structure (shades of teal) to wild-type Clr4 (yellow) (PDBID:6BOX) indicate absence of major differences in global

*Figure 3 continued on next page*

*Figure 3 continued*

folding. Residues 243–262 are colored in blue. Mutated residues (orange) and SAH co-factor are shown in stick representation. (**E**) Superposition of mutated region in stick representation for Clr4-3FA (orange) and Clr4 (gray) (PDBID:6BOX). (**F**) Methyltransferase kinetics of wild-type and Clr4KMT mutants measured by detection of $^3$H incorporation on SDS-PAGE gels shows loss of stimulation by H3K14ub for GS253 and 3FA mutants. To observe sufficient signal, differing enzyme concentrations of 20 and 200 nM were used for H3K14ub and H3 peptides, respectively. (**G**) Michaelis–Menten kinetics for Clr4 wild-type, 3FA, and GS253 mutants measured using the Promega MTase-Glo methyltransferase assay. Error bars correspond to standard error of mean based on three or four measurements.

The online version of this article includes the following figure supplement(s) for figure 3:

**Figure supplement 1.** Clr4F3A folds similarly to Clr4.

H3K14ub under the same conditions (*Figure 3B and C*). We also attempted to determine the $K_d$ for unmodified H3 peptides, but were unable to observe binding. The ITC data confirm that H3K14ub binds with high affinity to the KMT domain of Clr4 and that the UBR mutants disrupt the Clr4-H3K14ub enzyme-substrate complex.

The Clr4-3FA mutant showed very stable biochemical behavior and crystallized readily. To determine the effect of the 3FA mutations on the folding of Clr4, we solved the structure by X-ray crystallography. These efforts resulted in a 2.46 Å structure of Clr4-3FA (*Table 2*). The packing of the molecules in these crystals is similar to the packing observed in the structure of the autoinhibited Clr4, despite a difference in space group (PDBID:6BOX) (*Iglesias et al., 2018*). When comparing with previous X-ray structures of Clr4, we observed no significant difference in global or local protein folding between the wild-type Clr4 KMT domains and the Clr4-3FA mutant (RMSD = 0.37 Å) (*Figure 3D and E*, *Figure 3— figure supplement 1*). This indicates that the Clr4-3FA mutations do not induce significant structural changes.

To establish the impact of the mutants on the methyltransferase activity, we performed enzymatic assays using $^3$H-SAM and gel-based read-out. These assays show that the mutants lose activity on H3K14ub when compared to wild-type Clr4KMT (*Figure 3F*). To establish that the UBR mutants specifically target H3K14ub-mediated stimulation of Clr4 but not activity on unmodified peptide, we determined Michaelis–Menten kinetics for the Clr4-GS253 and Clr4-3FA mutants using a luminescence-based assay (*Figure 3G* , *Table 1*). Comparison of the specificity constants $k_{sp}$ shows that the efficiency of the UBR mutants on H3K14ub drops approximately 27-fold for Clr4-3FA and 48-fold for Clr4-GS253 compared to the wild-type. This drop in activity is caused by a corresponding drop in $K_M$, consistent with the ITC results. The difference between Clr4-3FA and Clr4-GS253 is not statistically significant, but is consistent with small differences observed in the radioactive gel-based and filter-binding assays (*Figure 3F*, *Figure 3—figure supplement 1C*).

In contrast to the enzymatic activity on H3K14ub, the activity on unmodified H3 peptides is only mildly affected resulting in a $k_{sp}$, that is, 91%  of wild-type for Clr4-3FA and 78%  for Clr4-GS253 (*Figure 3G*, *Table 1*). We conclude that the enzymatic and affinity measurements establish that both UBR mutants target the Clr4-H3K14ub interaction with high specificity.

## Clr4 loss-of-function mutants affect heterochromatin

In *S. pombe*, heterochromatin formation depends strongly on the H3K9me2/3 methyltransferase activity of Clr4. Clr4 knock-out and mutant strains are defective for transcriptional gene silencing and display hyperacetylated heterochromatic regions (*Nakayama et al., 2001*; *Bannister et al., 2001*; *Gerace et al., 2010*; *Iglesias et al., 2018*). The UBR mutants, therefore, provide a unique opportunity to investigate the functional importance of the H3K14ub-mediated stimulation for the *S. pombe* heterochromatin system. We introduced the Clr4-GS253 and Clr4-3FA mutations at the endogenous *clr4* locus and crossed them into a dual reporter (*imr1L::ura4*⁺/*otr1R::ade6*⁺) background for evaluating heterochromatin silencing using comparative growth assays (*Ekwall et al., 1997*). Strains with a functional silencing machinery are able to repress transcripts from these two centromeric reporters, and wild-type cells therefore show restricted growth on medium lacking uracil and turn red on medium containing low concentrations of adenine. Elevated levels of *ura4*, however, render the strains sensitive to growth on media containing 5-fluoroorotic acid (FOA), which is converted into the toxic product fluorodeoxyuridine when *ura4* is expressed at elevated levels. The *clr4-GS253* mutant showed increased growth on medium lacking uracil, no growth on FOA plates, and white colonies on low adenine, comparable to the *clr4Δ* mutant, which indicates a severe loss of transcriptional gene

**Table 2.** Crystallographic table.

| | Native |
|---|---|
| **Data** | |
| Wavelength (Å) | 0.97950 |
| Resolution range (Å) | 70.68–2.46 (2.68–2.46) |
| Space group | P 21 21 2 |
| Unit-cell parameters (Å, °) | 92.44, 110.29, 70.68, 90.00, 90.00, 90.00 |
| Total reflections | 120,264 (3409) |
| Unique reflections | 19,571 (979) |
| Multiplicity | 6.1 (3.5) |
| Completeness (%) spherical | 72.2 (16.2) |
| Completeness (%) ellipsoidal | 93.1 (56.9) |
| Mean $I/\sigma(I)$ | 6.6 (1.4) |
| Wilson B factor (Å$^2$) | |
| $R_{merge}$ | 0.214 (0.898) |
| $R_{meas}$ | 0.233 (1.046) |
| $R_{pim}$ | 0.093 (0.525) |
| CC1/2 | 0.993 (0.627) |
| CC | 0.998 (0.878) |
| **Refinement** | |
| Resolution range | 2.46 |
| Total number of reflections | 18,561 |
| Number of reflections in test set | 973 |
| $R_{work}$ (%) | 23.8 |
| $R_{free}$ (%) | 24.9 |
| CC (work) | 0.923 |
| CC (free) | 0.910 |
| No. of non-hydrogen atoms | 4495 |
| Macromolecule | 4411 |
| Ligands | 62 |
| Solvent | 22 |
| No. of protein residues | 549 |
| R.m.s.d., bonds (Å) | 0.014 |
| R.m.s.d., angles (°) | 1.73 |
| Ramachandran favored (%) | 96.54 |
| Ramachandran outliers (%) | 0 |
| Ramachandran allowed (%) | 3.46 |
| Clash score | 2.42 |
| Average B factor (Å$^2$) | |
| Macromolecule | 31.85 |
| Ligands | 24.61 |

*Table 2 continued on next page*

*Table 2 continued*

| | Native |
|---|---|
| Solvent | 21.63 |

Statistics for the highest resolution shell are shown in parentheses.

silencing (*Figure 4A*). The *clr4-3FA* mutant also showed a growth phenotype very similar to *clr4Δ*. However, the slightly pinkish color on low Ade plates suggests that the silencing defect is less severe than in *clr4-GS253* or *clr4Δ*. The *clr4* mutants are expressed at normal levels, and co-IP experiments with Rik1 show that they remain associated with the CLRC complex (*Figure 4B and C*). To get a quantitative measure of the loss of transcriptional gene silencing, we analyzed endogenous heterochromatic transcripts at centromeric *dg/dh* repeats and the subtelomeric *tlh1* locus by RT-qPCR (*Figure 4D*). The levels of these transcripts were greatly elevated in the *clr4-GS253* mutant, similar to *clr4Δ*. In contrast, the *3FA* mutant showed a more nuanced silencing defect with high transcript levels for centromeric *dh* and telomeric transcripts, and a modest ~10 -fold increase for centromeric *dg*. These findings are consistent with the weaker silencing defect of *clr4-3FA* observed in the growth assays and with the weaker loss of enzymatic stimulation in vitro. To test if the loss of gene silencing is associated with loss of H3K9 methylation, we performed chromatin immunoprecipitation (ChIP) against H3K9me1, H3K9me2, and H3K9me3 (*Figure 4E and F*). Both H3K9me2 and H3K9me3 were completely abolished in *clr4-GS253* and *clr4-3FA* strains at centromeric *dg/dh* repeats, being indistinguishable from *clr4Δ*. Currently, we cannot explain why the *dg* transcripts in the *3FA* mutant are only slightly elevated while completely losing H3K9me2/3. *clr4-GS253* and *clr4Δ* also showed elevated levels of H3K9me1 when compared to wild-type *clr4+*, while this was not observed for *clr4-3FA*. The elevated H3K9me1 marks support the hypothesis that another unidentified methyltransferase could be depositing this mark (*Jih et al., 2017*). We further found that in agreement with the increased transcript levels RNA polymerase II occupancy at *dh* repeats was strongly increased.

In summary, we observe defects in transcriptional gene silencing and heterochromatin formation for the UBR mutants that are very similar to *clr4Δ*. Furthermore, the severity of the phenotype correlates with the degree of loss of enzymatic function on H3K14ub in vitro, consistent with Clr4's KMT domain mediating the crosstalk between H3K14ub and H3K9me2/3 as an essential step in heterochromatin formation and maintenance.

## H3K14ub stimulation is conserved in mammalian SUV39H2

To investigate if stimulation of H3K9 methylation by H3K14ub is a conserved process, we purified the KMT domains of both human G9a and SUV39H2, as well as plant SUVH4/KRYPTONITE and performed methyltransferase assays to investigate the substrate preference of these enzymes by radiometric and fluorescence-based assays (*Figure 5A*, *Figure 5—figure supplement 1B and C*). SUV39H1 and SUV39H2 are the closest human homologs of Clr4 and are tightly associated with constitutive heterochromatin. In contrast, G9a regulates H3K9me2/3 deposition in euchromatin and directly regulates gene expression. SUVH4 is one of 10 SUVH genes in *Arabidopsis* and is involved in the maintenance of DNA methylation. Of our candidate enzymes, SUV39H2, consistently showed stimulation by H3K14ub, while we could not detect stimulation for G9a or SUVH4. These results suggest that the H3K14ub-mediated stimulation is conserved in a subset of SUV39 family proteins in higher eukaryotes, and that H3K14ub is potentially implicated in the regulation of H3K9 methylation in mammalian heterochromatin.

## Discussion

Histone methylation and acetylation are key post-translational modifications involved in the regulation of chromatin accessibility and transcription. We show here that the poorly understood histone modification H3K14ub leads to the strong stimulation of the methyltransferase activity of the fission yeast SUV39 family enzyme Clr4 through a sensing mechanism in its catalytic domain. The H3K14ub modification has, therefore, the biochemical potential to direct H3K9 methylation by Clr4 with high specificity. These results support a recently proposed regulatory role for H3K14ub in heterochromatin

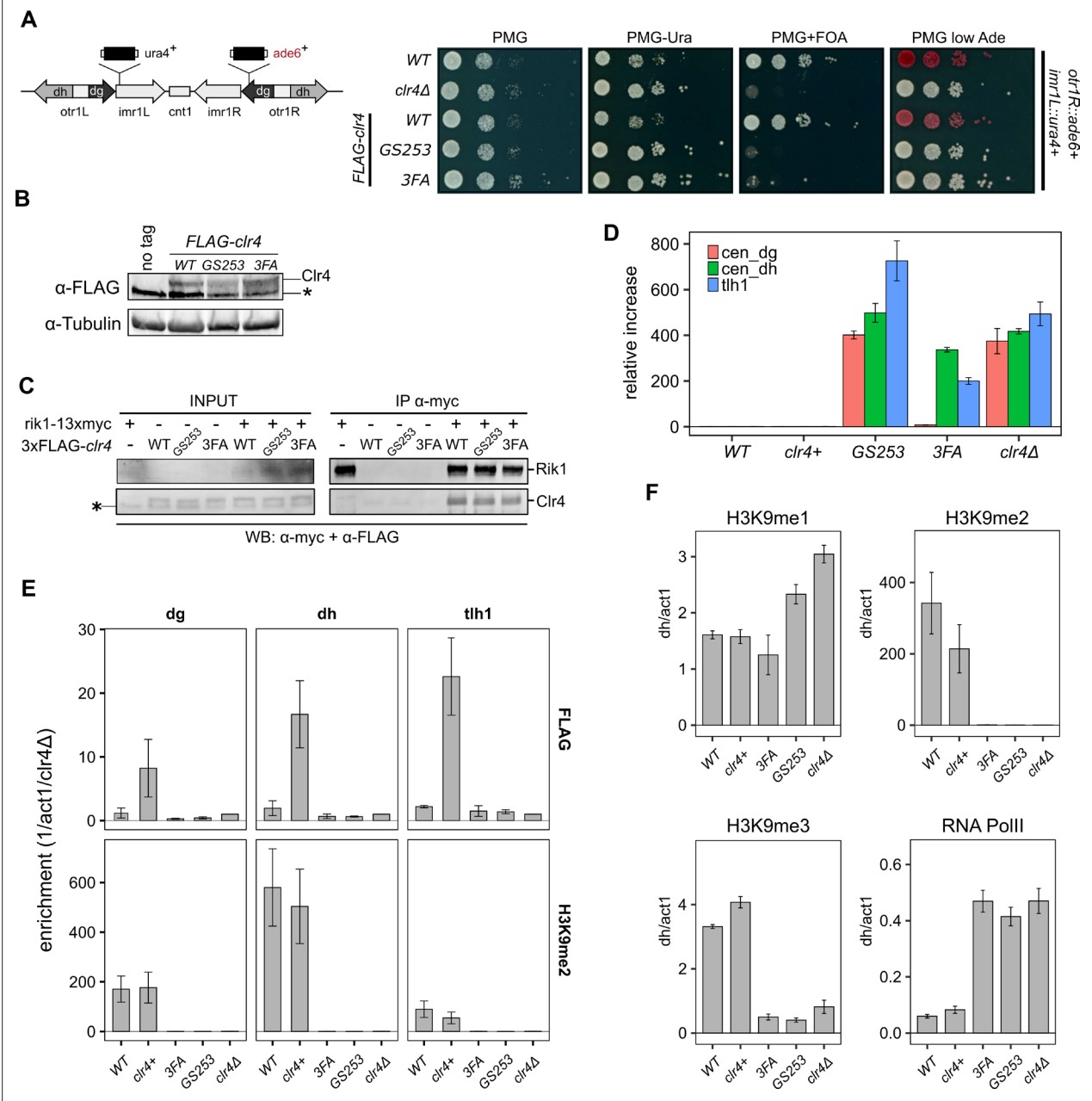

**Figure 4.** Clr4 mutants lose heterochromatin. (**A**) Serial dilution growth assays of wild-type and the Clr4 mutants. Strains were assessed for growth on PMG media, PMG-ura to monitor *imr1L::ura4+* expression, and PMG+FOA to monitor silencing of *imr1L::ura4+*. Expression of *otr1R::ade6+* was tested on PMG containing low adenine. (**B**) Immunoblot for FLAG-Clr4 on cell lysates from *clr4* mutant strains. Asterisk indicates a non-specific band. (**C**) Co-IP experiment to assess the stability of the CLRC complex in *clr4-GS253* and *clr4-3FA* mutants. (**D**) Changes in steady-state transcript levels in *clr4* mutant strains relative to wild-type cells were measured by RT-qPCR for centromeric *dg*, *dh* repeats, and *tlh1* transcripts at telomeres. *act1* was used as an internal standard for all measurements. (**E**) ChIP for wild-type and indicated mutant strains against FLAG-Clr4 and H3K9me2 at centromeric *dg*, *dh* repeats, and telomeric *tlh1*. Enrichment was normalized to *clr4Δ*. (**F**) ChIP for H3K9me1, H3K9me2 and H3K9me3, and RNA polymerase II at centromeric *dh* repeats. *act1* was used as an internal standard for all measurements. Mean and standard errors in (**D–F**) were calculated from a minimum of three independent biological replicates.

formation (*Oya et al., 2019*). Based on these and our findings, we propose that H3K14ub serves as a licensing mechanism for the deposition of H3K9me2/3 marks and heterochromatin formation (*Figure 5B*). In this model, acetylation of H3K14, which is strongly associated with euchromatin, assumes a much more specific role than normally associated with histone acetylation: it serves to prevent ubiquitination of H3K14 similar to a protecting group in organic chemical synthesis. The

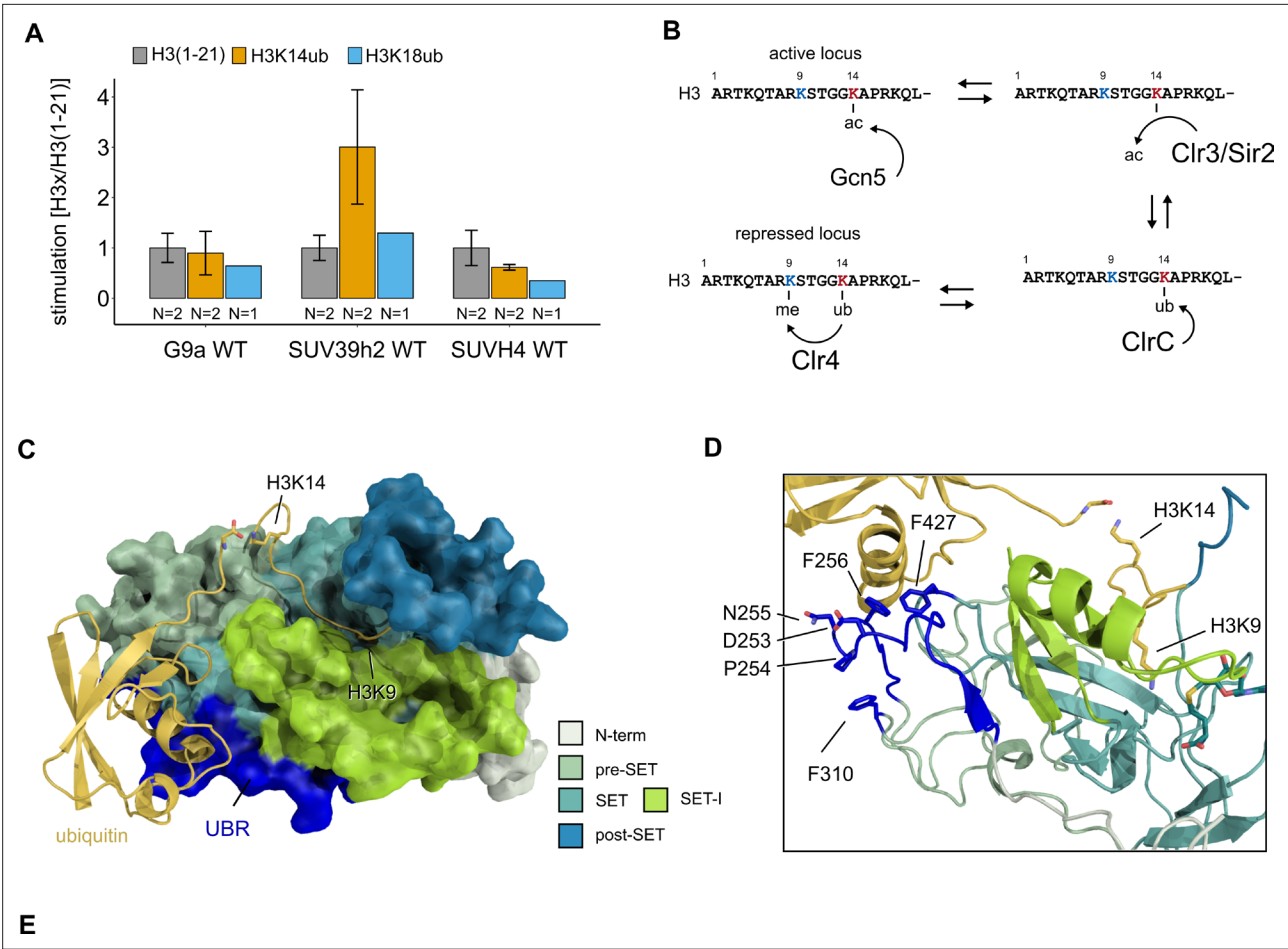

**Figure 5.** Conservation and mechanistic model of H3K14ub stimulation in the SUV39 methyltransferase family. (**A**) Methyltransferase rates of human G9a and human SUV39H2 and *Arabidopsis* SUVH4 on indicated substrates were determined by quantifying $^3$H-methyl incorporation using filter binding assays. Error bars correspond to standard error of mean. (**B**) Schematic of how H3K14 ubiquitylation licenses H3K9me2/3 deposition and how H3K14ac specifically prevents this. (**C, D**) Overview and detail view of the Clr4-H3K14ub model calculated with HADDOCK using restraints obtained by HDX-MS and mutagenesis.

The online version of this article includes the following figure supplement(s) for figure 5:

**Figure supplement 1.** Conservation and mechanistic model of H3K14ub stimulation in the SUV39 methyltransferase family.

H3K14ac-specific HDACs Clr3 and Sir2 therefore indirectly control H3K9me2/3 by regulating the availability of H3K14 for ubiquitination by the CLRC ubiquitin ligase complex. Consistent with this model, mutations of H3K14 lead to loss of H3K9 methylation (*Mellone et al., 2003*), which is phenocopied by double deletion of Clr3 and Sir2 (*Alper et al., 2013*). The genetic data further suggests that ubiquitination is essential for the establishment of heterochromatin since mutants of the CLRC complex are completely devoid of heterochromatin (*Horn et al., 2005*; *Jia et al., 2005*; *Li et al., 2005*; *Nakayama et al., 2001*; *Thon et al., 2005*). In the *clr4-GS253* and *clr4-3FA* mutants, we observe a severe loss of heterochromatin, which validates that recognition of H3K14ub by Clr4 is critically important for the establishment and maintenance of heterochromatin. In a parallel study, Shan et al. have shown that H3K14ub and the Clr4 UBR are critical factors for sequestration of Clr4 on H3K9M mutants, which mimic oncogenic histone lysine-to-methionine mutants (*Shan et al., 2021*).

We have consistently observed H3K14ub-mediated stimulation in reactions with the isolated KMT domain of Clr4, and we have shown that the N-terminus of Clr4 is not essential for recognition of H3K14ub. Our observations are difficult to reconcile with the results reported in *Oya et al., 2019*, where an intact N-terminus was required for stimulation. We speculate that this discrepancy is caused by the use of different protein expression constructs. While Oya et al. use GST-tagged proteins, we have removed the tags for all constructs. An influence on biochemical activity by the GST-tag has been

observed previously for SUV39H2 (*Schuhmacher et al., 2015*), and it is therefore conceivable that a GST-tag also influences the behavior of Clr4. Shan et al. have confirmed independently that H3K14ub stimulates the isolated KMT domain as observed here (*Shan et al., 2021*).

The HDX-MS and mutagenesis results provide a valuable set of constraints that we employed to predict the atomic structure of the Clr4-H3K14ub complex by protein-protein docking using HADDOCK (*Dominguez et al., 2003*). The resulting model (*Figure 5C*) shows that the H3K14ub peptide can comfortably fit the H3 peptide into the binding groove and simultaneously cover Clr4 residues 253–256 mutated in Clr4-GS253 and F427 mutated in Clr4-3FA with its K14-linked ubiquitin moiety. The model suggests that F310, which is also mutated in Clr4-3FA, is unlikely to be involved directly in the interaction with ubiquitin as the N-terminal residues 253–256 shield F310 from interacting with ubiquitin (*Figure 5D*).

Comparing this structural model with the best understood regulatory mechanisms of other SET domain proteins reveals interesting parallels and suggests a mechanism for how H3K14ub might stimulate the enzymatic activity of Clr4. EZH2, the catalytic subunit of the polycomb repressive complex PRC2, which methylates H3K27, is stimulated through conformational changes triggered by binding of its own H3K27me product to the neighboring subunit EED in the PRC2 complex (*Jiao and Liu, 2015*; *Justin et al., 2016*; *Brooun et al., 2016*). Intriguingly, the UBR that we identified by HDX-MS corresponds structurally to the 'SET activation loop' (SAL) observed in PRC2 structures, which stabilizes the SET-I domain, a region inside the SET domain that provides one side of the peptide binding groove (*Qian and Zhou, 2006*; *Wu et al., 2010*). In a further example, the mixed lineage leukemia complexes with catalytic SET domain proteins MLL1 or MLL3 are activated by structural rearrangement of the regulatory subunits RBBP5 and ASH2L. RBBP5 thereby assumes a similar position as the SAL in PRC2 and stabilizes the SET-I domain in coordination with ASH2L (*Li et al., 2016*). In these experiments, apo MLL3 and the MLL3-ASH2L-RBBP5 complex showed little structural differences in the positioning of SET-I, but complex formation significantly reduced the conformational dynamics of the SET-I domain. The structural similarities between these regulatory mechanisms identify the SET-I domain as a potential mediator of the H3K14ub stimulation. In our model, the SET-I domain is sandwiched between the H3K14-linked ubiquitin moiety and the H3 peptide bound in the active site (*Figure 5C and D*). Therefore, binding of H3K14ub to Clr4 might not only contribute to high substrate affinity but also to an optimally organized active site.

Bioinformatic analysis of the UBR sequence using Hidden Markov Models suggests that the UBR sequence is well conserved in the *Ascomycetes* clade of fungi, which includes the *Neurospora crassa* Dim-5 protein for example (*Figure 5—figure supplement 1D and E*). Comparing the Clr4 motif with motifs obtained using homologous sequences from human SUV39H2, human G9a, and *Arabidopsis* SUVH4 shows that SUV39H2's motif is similar to Clr4, while G9a and SUVH4 diverge significantly. This is consistent with our observation that H3K14ub can stimulate SUV39H2, but not G9A or SUVH4. The function of H3K14ub in human cells is poorly understood, but it has been detected by mass spectrometry (*Harrison et al., 2016*). In combination with the conservation of the H3K14ub mediated stimulation in SUV39H2, this suggests that the crosstalk observed in fission yeast is conserved in mammals. Further studies will be required to determine the prevalence of the crosstalk between H3K9me2/3 and H3K14ub, and the role it plays in the biology of eukaryotic organisms.

## Materials and methods

### Key resources table

| Reagent type (species) or resource | Designation | Source or reference | Identifiers | Additional information |
|---|---|---|---|---|
| Gene (*S. pombe*) | *clr4* | PomBase | SPBC428.08c | |
| Gene (*S. pombe*) | *rik1* | PomBase | SPCC11E10.08 | |
| Strain, strain background (*S. pombe*) | h + leu1-32 ura4DS/E ade6-M210 imr1R(NcoI)::ura4 ori1 | *Allshire et al., 1995* | FY498 | |
| Strain, strain background (*S. pombe*) | h- mat1m::cyhS smtO rpl42-P56Q (cyhR) ade6M210 leu1-32 ura4-D18 | *Roguev et al., 2007* | P392 | |

*Continued on next page*

*Continued*

| Reagent type (species) or resource | Designation | Source or reference | Identifiers | Additional information |
|---|---|---|---|---|
| Strain, strain background (*S. pombe*) | h− ade6-M210 his1-102 leu1-32 ura4-D18 otr1 R(dg-glu)Sph1::ade6 imr1L(Nco1)::ura4 | *Ekwall et al., 1997* | FY1191 | |
| Strain, strain background (*S. pombe*) | h + ade6 DN/N leu1-32 ura4-DS/E imr1L(Nco1)::ura4 otr1R(Sph1)::ade6 | *Ekwall et al., 1997* | FY2002 | |
| Strain, strain background (*S. pombe*) | h- ura4-D18 | Simanis Lab | S.057 | *Figure 4D* and E |
| Strain, strain background (*S. pombe*) | h- 3xFLAG-Clr4 imrlR(NcoI)::ura4 ade6-M210 leu1-32 ura4DS/E oril | This study* | S.0AT | |
| Strain, strain background (*S. pombe*) | h + rpl42-P56Q (cyhR) ade6M210 ura4-D18 clr4Δ::rpl42-natMX | This study* | S.0CT | - |
| Strain, strain background (*S. pombe*) | h + rpl42-P56Q (cyhR) ade6M210 ura4-D18 clr4Δ::KanMX | This study* | S.0D7 | - |
| Strain, strain background (*S. pombe*) | h- clr4Δ::KanMX ura4-D18 | This study* | S.0H1 | *Figure 4D* andE |
| Strain, strain background (*S. pombe*) | h- rik1::rik1-13myc-KanR | This study* | S.0JE | *Figure 4C* |
| Strain, strain background (*S. pombe*) | h + rik1::rik1-13myc-KanR ura4-D18 his2-? leu1-32 | This study* | S.0JL | - |
| Strain, strain background (*S. pombe*) | h- 3xFLAG-clr4-D253G/P254G/N255S/ F256G imr1R(Ncol)::ura4 oril ade6-M210 leu1-32 ura4DS/E | This study* | S.0KK | - |
| Strain, strain background (*S. pombe*) | h- 3xFLAG-Clr4 ade6-M210 leu1-32 ura4-D18 | This study* | S.0LL | *Figure 4C, D and E* |
| Strain, strain background (*S. pombe*) | h- 3xFLAG-clr4-D253G/P254G/N255S/ F256G ade6-M210 ura4-D18 | This study* | S.0LN | *Figure 4C, D and E* |
| Strain, strain background (*S. pombe*) | h- 3xFLAG-Clr4 imr1L(Nco1)::ura4 otr1R(Sph1)::ade6 ade6-? leu1-32 ura4-DS/E | This study* | S.0LP | *Figure 4A and B* |
| Strain, strain background (*S. pombe*) | h- 3xFLAG-clr4-D253G/P254G/N255S/ F256G imr1L(Nco1)::ura4 otr1R(Sph1)::ade6 ade6-? leu1-32 ura4-DS/E | This study* | S.0LS | *Figure 4A and B* |
| Strain, strain background (*S. pombe*) | h + clr4Δ::KanMX ade6-DN/N ura4-? | This study * | S.0MS | - |
| Strain, strain background (*S. pombe*) | h + rik1::rik1-13myc-KanR 3xFLAG-Clr4 his2-? leu1-32 ura4-D18 | This study* | S.0MT | *Figure 4C* |
| Strain, strain background (*S. pombe*) | h− rik1-13myc 3xFLAG-clr4-D253G/P254G/ N255S/F256G his2-? leu1-32 ura4-D18 | This study* | S.0MU | *Figure 4C* |
| Strain, strain background (*S. pombe*) | h- 3xFLAG-clr4-F256A/F310A/F427A rpl42-P56Q (cyhR) ade6M210 ura4-D18 | This study* | S.0MX | *Figure 4C* |
| Strain, strain background (*S. pombe*) | h- clr4Δ::KanMX otr1R(dg-glu)Sph1::ade6 imr1L(Nco1)::ura4 ade6-? leu1-32 ura4-? | This study* | S.0NC | *Figure 4A* |
| Strain, strain background (*S. pombe*) | h- 3xFLAG-clr4-F256A/F310A/F427A otr1R(dg-glu)Sph1::ade6 imr1L(Nco1)::ura4 ade6-M210 ura4-D18 | This study* | S.0ND | *Figure 4A and B* |
| Strain, strain background (*S. pombe*) | h- 3xFLAG-clr4-F256A/F310A/F427A rik1::rik1-13myc-KanR ade6-M210 ura4-D18 | This study* | S.0NF | *Figure 4C, D and E* |
| Sequence-based reagent | Oligonucleotides | *Supplementary file 1A* | | |

*Continued on next page*

*Continued*

| Reagent type (species) or resource | Designation | Source or reference | Identifiers | Additional information |
|---|---|---|---|---|
| Sequence-based reagent | Peptides | *Supplementary file 1B* | | |
| Sequence-based reagent | sgRNAs | *Supplementary file 1C* | | |
| Recombinant DNA reagent | pSMT3 (6xHIS tag) | Chris Lima – Cornell University *Mossessova and Lima, 2000* | | 6xHIS-Sumo tagging plasmid |
| Recombinant DNA reagent | pSMT3_Clr4(1-490) | This study* | P.0PI | *Figure 1* |
| Recombinant DNA reagent | pSMT3_Clr4KMT (192-490) | This study* | P.0QW | *Figures 1 and 2*, and 3 |
| Recombinant DNA reagent | pSMT3_Clr4KMT (192-490)-GS253 | This study* | P.18B | *Figure 3* |
| Recombinant DNA reagent | pSMT3_Clr4KMT (192-490)–3FA | This study* | P.18C | *Figure 3* |
| Recombinant DNA reagent | pSumo-RSFDuet SUVH4 (93–624) | Steven Jacobsen – University of California Los Angeles *Du et al., 2014* | | *Figure 5* |
| Recombinant DNA reagent | pet28a_SUV39H2 | Addgene, Cheryl Arrowsmith | RRID:Addgene_25115 | *Figure 5* |
| Recombinant DNA reagent | pET38a_G9a | Addgene, Cheryl Arrowsmith | RRID:Addgene_25503 | *Figure 5* |
| Antibody | Rabbit polyclonal anti-H3K9me1 | Abcam | ab8896 | (ChIP: 1 µg) |
| Antibody | Mouse monoclonal anti-H3K9me2 | Abcam | ab1220 | (ChIP: 1 µg) |
| Antibody | Recombinant mono clonal anti-H3K9me3 | Diagenode | C15500003 | (ChIP: 1 µg) |
| Antibody | Mouse monoclonal anti-RNA PolII | Abcam | ab817 | (ChIP: 1 µg) |
| Antibody | Mouse monoclonal anti-FLAG-M2 | Sigma | F1804 | (ChIP: 1 µg, WB: 1:5000) |
| Antibody | Rabbit recombinant monoclonal anti-H3K14ac | Abcam | ab52946 | (ChIP: 1 µg) |
| Antibody | Mouse monoclonal anti-Myc-Tag | Cell Signaling Technology | 9B11 | (WB: 1:3000) |
| Antibody | Mouse monoclonal anti-γ-Tubulin | Sigma-Aldrich | T6557 | (WB: 1:5000) |
| Chemical compound, drug | 5-Fluoroorotic acid (FOA) | US Biological | F5050 | |
| Chemical compound, drug | Geneticin (G418 sulfate) | Invitrogen | 10131019 | |
| Chemical compound, drug | Cycloheximide | Alfa Aesar | J66901.03 | |
| Chemical compound, drug | Nourseothricin | Werner Bioagents | 5.001.000 | |
| Chemical compound, drug | Formaldehyde | Sigma-Aldrich | F8775 | |
| Chemical compound, drug | Ethylene glycol bis-succinimidyl succinate (EGS) | Thermo Fisher Scientific | 21565 | |

*Continued on next page*

*Continued*

| Reagent type (species) or resource | Designation | Source or reference | Identifiers | Additional information |
|---|---|---|---|---|
| Chemical compound, drug | Dynabeads Protein A | Thermo Fisher Scientific | 10001D | |
| Chemical compound, drug | Dynabeads MyOne Streptavidin C1 | Thermo Fisher Scientific | 65001 | |
| Chemical compound, drug | Trizol | Thermo Fisher Scientific | 15596026 | |
| Chemical compound, drug | cOmplete EDTA free | Roche | 11873580001 | |
| Chemical compound, drug | Myc-Trap | Chromotek | ytma-20 | |
| Chemical compound, drug | Adenosyl-L-methionine, S-[methyl-3H]/SAM | Perkin Elmer | NET155V250UC | |
| Chemical compound, drug | Phosphocellulose paper 541 | Jon Oakhill, St Vincent's Institute of Medical Research, Melbourne, Australia | | |
| Commercial assay or kit | EvoScript Universal cDNA Master | Roche | 07912439001 | |
| Commercial assay or kit | LightCycler 480 SYBR Green I Master | Roche | 04707516001 | |
| Commercial assay or kit | EPIgeneous Methyltransferase Assay kit | Cisbio | 62SAHPEB | |
| Commercial assay or kit | MTase-Glo Methyltransferase Assay | Promega | V7601 | |
| Software, algorithm | RStudio | RStudio, Inc | Version 1.2.5042 | Fitting enzyme kinetics and plotting graphs |
| Software, algorithm | R | R Foundation | Version 3.6.3 | Fitting enzyme kinetics and plotting graphs |

*Reagent is available upon request from the authors.

## Generation of *S. pombe* strains

*S. pombe* strains were grown and manipulated using standard techniques unless differently stated. The list of strains used in this study can be found in the Key resources table. Strains were crossed and analyzed either by tetrad dissection or random spore analysis. For tetrad dissection, only spores from complete tetrads were further analyzed. In the absence of any selection markers, the genotype was analyzed by colony PCR. The strain *clr4Δ::kanMX* was generated by transforming a wild-type strain with a DNA fragment containing the *kanMX* cassette plus 80 bp of sequence found upstream and downstream of *clr4* ORF. The transformants were selected on YES plates containing 100 µg/ml of G418, and correct integration was confirmed by PCR and sequencing. The strain *3xFLAG-clr4* was generated by CRISPR mutagenesis of a wild-type strain using as homologous repair template a fragment of DNA containing the 3xFLAG tag plus 500 bp of homology flanking the cleavage site and a silent point mutation disrupting the PAM site. The *3xFLAG-clr4-GS253* strain was generated with the same approach by transforming the *3xFLAG-clr4* strain with DNA containing the desired mutations and homologies. The sgRNA sequences used for CRISPR are indicated in *Supplementary file 1C*. The *clr4Δ::rpl42-natMX* strain was generated by transformation of a *rpl42.sP56Q* host strain with a DNA fragment containing the *rpl42-natMX* cassette plus 500 bp of homology for sequences flanking *clr4* ORF. The integrants were selected by rounds of positive selection on rich media containing 100 µg/ml nourseothricin, and followed by negative selection on plates containing 100 µg/ml cycloheximide. The integrations were tested by PCR and sequencing. To generate the *3xFLAG-clr4-3FA* mutant strain, the *clr4Δ::rpl42-natMX* strain was transformed with a DNA fragment containing the desired mutations and

500 bp homology arms outside the *rpl42-natMX* cassette. The strain was selected with rounds of positive and negative selection on YES plates containing cycloheximide and nourseothricin, respectively. The strain *rik1-13xmyc-kanR* was generated by transformation of a wild-type strain with a *13xmyc-kanR* cassette with homology for the 3'end of *rik1*. The strain was selected on plates containing G418. All transformants were analyzed by PCR, sequenced, and the strains were outcrossed at least once. For plate-based silencing assays, reporter strains were generated by crossing the strains of interest with strains containing the marker genes *ura4+* and *ade6+* integrated at the centromere I (*imr1L::ura4+*, *otr1R::ade6+*).

## Silencing assays in *S. pombe*

Strains were grown overnight at 30 °C in YES to a density of $5 \times 10^6$ cells/ml. Cells were harvested by centrifugation and resuspended in water. 10-fold serial dilutions were spotted onto PMG, PMG-Ura, and PMG +1 mg/ml 5-FOA, with $1 \times 10^4$ cells in the highest density spots.

## RT-qPCR

RT-qPCR was performed as previously described (*Leopold et al., 2019*). Briefly, strains were grown in YES to a density of $5 \times 10^6$ cells/ml, harvested by centrifugation, washed with water, and frozen at –70 °C until further usage. RNA was purified by Trizol extraction followed by phenol/chloroform extraction and ethanol precipitation. Contaminant DNA was removed by DNaseI treatment and phenol/chloroform extraction followed by ethanol precipitation. cDNA was generated with Evo-Script Universal cDNA Master kit (Roche) and qPCR performed using SYBR Green I Master kit on LightCycler 480 instrument (Roche). qPCR primers used in this study are listed in *Supplementary file 1A*, and *act1+* was used for normalization. Data were analyzed using the ΔΔCt method (*Yuan et al., 2006*).

## Chromatin immunoprecipitation (ChIP)

ChIP was performed as previously described with some modifications (*Leopold et al., 2019*). Briefly, cells for ChIP were grown in 50 ml of YES medium to a density of $2 \times 10^7$ cells/ml. For RNA Polymerase II, H3K9me1, H3K9me2, and H3K9me3, cells were fixed in 1% formaldehyde for 30 min. For 3xFLAG-Clr4, a dual-crosslinking approach was employed as previously described (*Tian et al., 2012*). Cells were incubated at 18 °C for 2 hr, resuspended in 5 ml PBS, and crosslinked at room temperature (RT) with 1.5 mM ethylene glycol bis-succinimidyl succinate (EGS, Thermo Scientific). After 30 min of incubation, 1% formaldehyde was added and cells were crosslinked for further 30 min. The residual formaldehyde was quenched with 125 mM glycine for 5 min. Cells were washed and stored at –70 °C until further usage.

Pellets were resuspended in ChIP Buffer (50 mM HEPES/KOH pH 7.6, 150 mM NaCl, 1 mM EDTA, 1% Triton X-100, 1 mM PMSF, cOmplete EDTA free (Roche)), and lysed by bead-beating. Chromatin was enriched by centrifugation, resuspended in lysis buffer, and sonicated for 15 min (30 s/30 s ON/OFF) in a Bioruptor Pico. Cell debris was removed by centrifugation. For each IP, 50 µl of sheared soluble chromatin was diluted with 450 µl ChIP Buffer, mixed with 1 µg of antibody and incubated for 2 hr, followed by 45 min incubation with 30 µl of Dynabeads Protein A or G. For H3K9me3, 0.75 µg of anti-H3K9me3 (Diagenode C15500003) was pre-incubated with 30 µl of Dynabeads MyOne Streptavidin C1, followed by blocking with 5 µM biotin, according to the manufacturer's instructions. The loaded beads were then added to the diluted chromatin and incubated for 2 hr.

The bead/protein complex was washed three times with ChIP Buffer, once with high salt buffer (50 mM HEPES/KOH pH 7.6, 500 mM NaCl, 1 mM EDTA), once with LiCl buffer (5 mM Tris-Cl pH 8, 250 mM LiCl, 0.5% Triton X-100, 0.05 % Tween 20), and once with TE (10 mM Tris-Cl pH 8, 1 mM EDTA). The protein-DNA complex was eluted in elution buffer (50 mM Tris-Cl pH 8, 10 mM EDTA, 1% SDS) at 65 °C for 15 min, and the crosslinking reversed overnight at 65 °C. The samples were then treated with proteinase K, and DNA was purified by phenol-chloroform. qPCR was performed using the primers given in *Supplementary file 1A*; *act1+* was used as internal control.

## Co-immunoprecipitation from *S. pombe* cells

Cells were grown overnight at 30 °C to a density of $1.0 \times 10^7$ cells/ml, washed once with PBS, and the pellet frozen at –70 °C until further usage. Pellets were resuspended in co-IP buffer (50 mM HEPES/KOH pH 7.5, 300 mM NaCl, 1 mM EDTA, 5 mM CHAPS, 10 mM DTT) supplemented with cOmplete

EDTA free (Roche). The cells were mixed with the same volume of 0.5 mm glass beads and ruptured by bead beating in a Fastprep24 (MP Biomedical). Cell debris was removed by centrifugation and the supernatant incubated with Myc-Trap magnetic beads (ChromTek) for 1 hr at 4 °C. The unbound fraction was removed, and the beads were washed five times with co-IP buffer. The bound proteins were eluted by boiling in PGLB (125 mM Tris-HCl, 4% SDS, 50% glycerol, 0.2% Orange G) supplemented with 100 mM DTT. The proteins of interest were analyzed by Western blot.

## Western blotting

Samples were run on 10% Bis-Tris gels and transferred to nitrocellulose membrane (Bio-Rad). Proteins were detected by Western blotting using antibodies against either tag or protein of interest, followed by incubation with secondary antibodies labeled with the DyLight fluorophores, and scanning with the Odyssey Imaging System (LI-COR).

## Protein expression in *Escherichia coli*

All Clr4 protein constructs were cloned as N-terminal fusions to a 6xHIS-SUMO-tag in pSMT3 using Gibson cloning (*Gibson et al., 2009*; *Mossessova and Lima, 2000*). All constructs were expressed in the Rosetta2(DE3) strain. Cells were grown at 37 °C to log-phase in 2xTY media supplemented with 2% ethanol, cooled down on ice, and induced with 0.4 mM IPTG at 16 °C for 18 hr. Pellets were resuspended in NiNTA wash buffer (50 mM Tris-HCl pH 8.0, 300 mM NaCl, 20 mM imidazole pH 7.5, 1 mM β-mercaptoethanol) supplemented with protease inhibitor cocktail (PIC) (2 µM pepstatin, 2 µM leupeptin, 1 mM PMSF, 1 mM Benzamidine, 1 µg/ml aprotinin) and frozen at –80 °C until further usage.

## Protein purification from *E. coli*

For purification, frozen cells were quickly thawed at 37 °C, supplemented with PIC, and ruptured by using either sonication or the Emulsiflex homogenizer. Cell debris was removed by centrifugation. The cleared lysate was filtered through a 0.45 µm filter and loaded on a HisTrap HP column (GE Healthcare). The full-length Clr4 was enriched from the cleared lysate by fractionation with 40% $NH_4SO_4$. The pellet containing Clr4 was then resuspended in NiNTA wash buffer (50 mM Tris-HCl pH 8.0, 300 mM NaCl, 20 mM imidazole pH 7.5, 1 mM β-mercaptoethanol), filtered through a 0.45 µm filter, and loaded on a HisTrap HP column. After washing with NiNTA wash buffer, the proteins were eluted by either gradient or step elution in elution buffer (50 mM Tris-HCl pH 8.0, 300 mM NaCl, 300 mM imidazole pH 7.5, 1 mM β-mercaptoethanol). All proteins containing a 6xHIS-SUMO-tag were treated with Ulp1 while being dialyzed overnight in cleavage buffer (5 mM Tris-HCl pH 8.0, 150 mM NaCl, 2 mM β-mercaptoethanol). SUV39H2 and G9a were treated with thrombin to remove the 6xHIS-tag and dialyzed into cleavage buffer. The cleaved tags and Ulp1 were removed by negative purification on a HisTrap HP column. All proteins were further purified by SEC (Superdex75 or Superdex200, GE Healthcare) in gel filtration buffer (5 mM Tris-HCl pH 8.0, 150 mM NaCl, 5 mM DTT). Peak fractions were concentrated as needed in Amicon Ultra spin concentrators. For methyltransferase assays, the concentrated proteins were mixed 1:1 with glycerol, frozen in liquid nitrogen, and stored at –8 0°C until further usage. Clr4FL, SUV39H2, and SUVH4 were frozen without glycerol.

## Materials for peptide synthesis

All solvents and reagents were purchased from commercial sources and used without further purification. All amino acid derivatives, 2-chlorotrityl chloride (2 Cl Trt) resin and 2-(7-Aza-1H-benzotriazole -1-yl)–1,1,3,3-tetramethyluronium hexafluorophosphate (HATU) were purchased from Novabiochem, Merck (Darmstadt, Germany). N,N-Dimethylformamide (DMF), N,N-diisopropylethylamine (DIEA), and piperidine were from Acros Organics (Geel, Belgium). O-(6-Chlorobenzotriazol-1-yl)-N,N,N',N'-tetramethyluronium hexafluorophosphate (HCTU) was from Carl Roth GmbH (Karsruhe, Netherlands). Hydrazine monohydrate was purchased from Alfa Aesar (Heysham, UK), acetonitrile (ACN) from Avantor Performance Materials (USA). Tentagel S RAM resin, hydroxybenzotriazole (HOBt), tris(2-carboxyethyl)phosphine (TCEP), diethylether, phenylsilane, tetrakis(triphenylphosphine)palladium(0), trifluoroacetic acid (TFA), dichloromethane (DCM), triisopropylsilane (TIS), L-glutathione reduced (GSH), sodium diethyldithiocarbamate trihydrate, and methyl thioglycolate (MTG) were from Sigma-Aldrich (Taufkirchen, Germany). 2,2'-Azobis[2-(2-imidazolin-2-yl)propane]dihydrochloride (VA-044) was purchased from Wako Pure Chemical Industries, Ltd. (Osaka, Japan). All other commonly used

chemical reagents and buffer components were from Applichem (Darmstadt, Germany) and Fisher Scientific (Reinach, Switzerland).

## Instrumentation for peptide synthesis and purification

Reaction vessels for manual peptide synthesis as well as the automated Tribute peptide synthesizer were from Protein Technologies Inc. Analytical RP-HPLC analysis was performed on an Agilent 1260 series instrument using an analytical Agilent Zorbax C18 column (column dimensions: 150 mm × 4.6 mm, 5 µm particle size) at a flow rate of 1 ml/min. All RP-HPLC analyses were done with 0.1% (v/v) TFA in $H_2O$ (RP-HPLC solvent A) and 90% ACN and 0.1% (v/v) TFA in $H_2O$ (RP-HPLC solvent B) as mobile phases. Typically, a gradient from 0% to 70% solvent A to solvent B over 30 min was used for analytical RP-HPLC analyses unless otherwise stated. Purification of proteins on a semi-preparative scale were performed on an Agilent 1260 series instrument using a semi-preparative Agilent Zorbax C18 column (column dimensions: 250 mm × 9.4 mm, 5 µm particle size) at a flow rate of 4 ml/min. Lyophilization was carried out with a Telstar LyoQuest freeze dryer. Electrospray ionization mass spectrometric (ESI-MS) analysis was conducted on a Shimadzu MS2020 single quadrupole instrument connected to a Nexera UHPLC system. Mass spectra were acquired by electrospray ionization in positive ion mode in the mass range of 200–2000 m/z.

## Preparation of Fmoc-Aa-hydrazine-Cl-trityl-resin

Preparation of Fmoc-Arg(Pbf)-NHNH-Cl-Trt-resin was performed as reported previously (*Stavropoulos et al., 1996*). In general, 0.50 g 2-Cl-Trt-resin (substitution: 1.63 mmol/g, 1.00 eq., 0.82 mmol) were swollen in 3.00 ml DMF for 30 min at RT. Subsequently, the resin was allowed to cool to 0 °C and 1.00 ml of a solution containing DIEA (3.00 eq., 2.45 mmol, 427 µl) and hydrazine monohydrate (2.00 eq., 1.64 mmol, 80 µl) in DMF was added dropwise. The reaction mixture was stirred 1 hr at RT. 100 µl methanol (MeOH) was added, the resin was stirred 10 min at RT, transferred to a reaction vessel for manual peptide synthesis, and washed thoroughly with DMF.

Due to the low stability of the hydrazine resin, the first amino acid Fmoc-Arg(Pbf)-OH was coupled manually by standard Fmoc chemistry (*Atherton and Sheppard, 1989*). The loading of the 2-Cl-Trt-resin with hydrazine was assumed to be 50% (1.00 eq., 0.41 mmol). Fmoc-Arg(Pbf)-OH (5.00 eq., 2.05 mmol) was activated with 3.90 ml of a 0.50 M HATU solution (4.76 eq., 1.95 mmol) in DMF followed by 2 min incubation at RT. Then, 714 µl DIEA (10.00 eq., 4.10 mmol) were added and the reaction mixture was incubated another 1 min at RT. The activated amino acid was added to the resin, incubated 30 min at RT, and washed with DMF. To ensure high yield, the coupling procedure was repeated. Finally, the resin was washed with DMF, DCM, and MeOH and dried under vacuum. Resin substitution was determined by treating a defined amount of resin with 20% piperidine in DMF for 30 min at RT, followed by spectrophotometric quantification of released dibenzofulven-related species (resin loading: 0.51 mmol/g).

## Automated solid phase peptide synthesis (SPPS)

The sequences of the two peptides synthesized by Fmoc-SPPS are summarized in *Supplementary file 1*B. The synthetic procedures and analytical data are presented in the following. General protocol: the peptides were synthesized by the Tribute peptide synthesizer (PTI) on the previously prepared Fmoc-Arg(Pbf)-hydrazine-Cl-Trt-resin (peptide P1) or on Tentagel S RAM resin (peptide P2) yielding peptides with C-terminal hydrazide (peptide P1) or amide (peptide P2). The syntheses were performed on 0.1 mmol scale using Fmoc chemistry. The following base-resistant groups were employed to protect amino acid side chains: Arg(Pbf), Thr(tBu), Lys(Boc), Gln(Trt), Asn(Trt), Ser(tBu). Of note, Lys14 (peptide P1) and Lys18 (peptide P2) were protected with Alloc protecting group. To maximize synthesis yield, pseudoproline dipeptide building blocks were used and amino acids were double coupled where necessary.

Briefly, the N-terminal Fmoc-group was deprotected with 20% (v/v) piperidine in DMF. Activation of amino acid (5.00 eq., 0.50 mmol) was achieved by addition of HCTU (4.76 eq. 0.48 mmol) and DIEA (10.00 eq., 1.00 mmol). The coupling step was performed by adding the activated amino acid to the resin, followed by 30 min incubation at RT. When the full-length peptides were assembled, the peptidyl-resin was washed with DMF, DCM, and MeOH and dried under vacuum. Importantly, the

N-terminal Fmoc protecting group was not removed at this stage to allow further manipulation of the peptide.

## Alloc deprotection

The peptidyl-resin was swollen for 30 min in DCM. Alloc deprotection of Lys14 (peptide P1) or Lys18 (peptide P2) was initiated by addition of 1.00 ml of dry DCM and PhSiH$_3$ (24 eq., 24 mmol), followed by Pd(PPh$_3$)$_4$ (0.25 eq., 0.025 mmol) in 3.00 ml dry DCM. The peptidyl-resin was incubated for 30 min at RT and washed with DCM. The deprotection reaction with PhSiH$_3$ and Pd(PPh$_3$)$_4$ was repeated two more times. The resin was thoroughly washed with DCM followed by washing with 0.5% (v/v) DIEA in DMF; 0.5% (w/v) sodium-diethyldithiocarbamate in DMF; 50% (v/v) DCM in DMF; 0.5% (w/v) HOBt in DMF and extensively washed with DMF.

## Manual coupling of Boc-Cys(Trt)-OH

The cysteine residue was coupled manually to the $\varepsilon$-amino group of Lys14 (peptide P1) or Lys18 (peptide P2), yielding peptides P1′ and P2′ respectively (*Figure 1—figure supplement 1A*). Boc-Cys(Trt)-OH (5 eq., 0.50 mmol) was activated by addition of 0.95 ml of a 0.50 M HATU solution (4.76 eq., 0.48 mmol) in DMF, followed by 2 min incubation at RT. 172 µl DIEA (10 eq., 1 mmol) were added and the reaction mixture was incubated another 1 min at RT. The activated amino acid was added to the peptidyl-resin, incubated 30 min at RT, and washed with DMF. To ensure high yield, the coupling procedure was repeated. Finally, the resin was washed thoroughly with DMF.

## N-terminal Fmoc deprotection and cleavage from the resin

Fmoc deprotection was achieved by treating the peptidyl-resin with 10.00 ml 20% (v/v) piperidine in DMF for 5 min. The deprotection step was repeated.

The peptides were cleaved from the resin using either 95% (v/v) TFA, 2.5% (v/v) TIS, and 2.5% (v/v) H$_2$O. The crude peptides were precipitated by addition of ice-cold diethyl ether, recovered by centrifugation, dissolved in 50% (v/v) acetonitrile in H$_2$O, flash-frozen, and lyophilized.

## Preparation of ubiquitin-MESNa

Ubiquitin (G76C) was expressed in Rosetta2(DE3) cells as a C-terminal fusion to the N-terminal half of the Npu split intein, followed by a hexahistidine tag (*Kilic et al., 2018*). Bacterial cells were resuspended in lysis buffer (50 mM Na-phosphate buffer pH 7.0, 300 mM NaCl, 40 mM imidazole pH 7.0) and ruptured by sonication. Ubiquitin-NpuN-6xHIS was purified by Ni-affinity chromatography and eluted with 300 mM imidazole. The ubiquitin thioester was generated by intein derivatization with 100 mM sodium 2-mercaptoethanesulfonate (MESNa). The reaction was allowed to proceed for 16 hr at RT. Imidazole was removed by diafiltration and ubiquitin-MESNa was further purified by negative Ni-affinity chromatography to remove the NpuN-6xHIS tag. Finally, ubiquitin-MESNa was diafiltrated against 0.1% TFA, lyophilized, and stored at –20 °C.

## One-pot ligation and desulfurization

Ubiquitin-MESNa (1.00 eq., 1.16 µmol) was dissolved in 115.80 µl ligation buffer (6.00 M GmdHCl, 0.20 M sodium phosphate, pH 7, degassed) to a final concentration of 10.00 mM and 0.77 µl MTG (7.50 eq., 8.70 µmol) were added to the solution. The thiol-thioester exchange reaction was allowed to proceed for 20 min at RT. Ligation between ubiquitin-MTG thioester and the cysteine residue coupled to Lys14 (peptide P1) or Lys18 (peptide P2) was initiated by addition of peptide P1′ or peptide P2′ (1.30 eq., 1.51 µmol), followed by addition of 5.80 µl of 0.50 M TCEP solution in ligation buffer. The ligation mixture was incubated at 25 °C for 16 hr. The progress of the reaction was monitored by RP-HPLC and ESI-MS analysis.

When the ligation was complete, radical desulfurization of the cysteine at the ligation site was performed in the same reaction tube without prior purification of the ligation product. TCEP desulfurization buffer (0.50 M TCEP, 6.00 M GdmHCl, 0.20 M phosphate, pH 7) was added to a final TCEP concentration of 0.25 M. The desulfurization reaction was initiated by addition of VA-044 and GSH to a final concentration of 30 mM and 40 mM, respectively. The reaction mixture was incubated at 42 °C for 6 hr, and the progress of the reaction was monitored by RP-HPLC and ESI-MS analysis, yielding ubiquitin adducts P1″ and P2″ (*Figure 1—figure supplement 1*).

### Purification of H3K14ub (P1″) and H3K18ub (P2″)

The ubiquitin adduct peptides P1″ and P2″ were purified by semi-preparative RP-HPLC on a linear gradient from 25% to 55% solvent B over 40 min. Pure fractions were pooled, lyophilized, and analyzed by analytical RP-HPLC and ESI-MS (*Figure 1—figure supplement 1B and C*).

### Tritium-based histone methyltransferase assay

The HMT assays were performed in a total volume of 10 µl by mixing the substrates with the indicated constructs in reaction buffer (25 mM sodium phosphate pH 8.0, 100 mM NaCl, 10% v/v glycerol, 2.9 µM (0.5 µCi) *S*-[Methyl-3 H]-adenosyl-L-methionine [1 mCi/mL, Perkin Elmer]) and incubated at 30 °C. Measurements were performed with 20 µM of substrate, and 20 nM or 200 nM of each enzyme. Assay-specific concentrations and incubation times are indicated in the text and captions. For gel-based assays, the reactions were stopped by addition of 5 µl of PGLB , 100 mM DTT and the unreacted SAM was removed by SDS-PAGE. The gels were stained with Coomassie Brilliant Blue, imaged, and incubated for 30 min in EN3HANCE (Perkin Elmer) prior to drying. The methylated histones were detected by fluorography.

For filter binding assays, the reactions were stopped by addition of 5 µl 75% acetic acid, and 10 µl of each mixture was spotted onto phosphocellulose filter paper disks. Each filter was washed for 5 min in 2 ml 100 mM sodium bicarbonate pH 9.0 for a total of three times. The activity on each disk was quantified in a liquid scintillation counter after 1 hr incubation in 3 ml ULTIMA GOLD F cocktail.

### TR-FRET-based methyltransferase assay

The assays were performed using the EPIgeneous Methyltransferase kit (Cisbio). The reactions were performed by mixing the Clr4 constructs (30 nM) with the indicated substrates (1 nM –-200 µM for *Figure 1D*, 2 µM in *Figure 1E*, and 20 µM in *Figure 5—figure supplement 1B*) in Mtase Buffer (25 mM sodium phosphate pH 8.0, 100 mM NaCl, 10% v/v glycerol) supplemented with 10 µM SAM and incubated at 30 °C for 15 min, unless differently stated. The reactions were stopped and measured using a TECAN SPARK plate reader. Detection was performed at 620 and 665 nm wavelength. The 665/620 signal ratio was used to extrapolate the concentration of SAH from a standard curve.

### Bioluminescence-based methyltransferase assay

Methyltransferase assays were performed with wild-type and mutant Clr4KMT domains on H3K14ub (using 20 nM Clr4) and unmodified H3 (using 1 µM Clr4) in Mtase Buffer supplemented with SAM (10 µM for H3K14ub, 100 µM for H3). Reactions were stopped by addition of TFA to 0.1% and the production of SAH was measured using the MTase-Glo Methyltransferase Assay kit (Promega) according to the manufacturer's instructions using a Hidex Sense microplate reader.

### Hydrogen deuterium exchange mass spectrometry (HDX-MS)

HDX reactions were done in 50 µl volumes with a final protein concentration of 3 µM of enzyme. The dynamics of Clr4 was compared in three different conditions: (i) soluble Clr4, (ii) complex of Clr4 bound to ubiquitinated H3 peptide, and (iii) Clr4 in the presence of an excess of H3 peptide. Dynamics of ubiquitinated H3 peptide was investigated by comparing its D2O incorporation alone or when in complex with Clr4. Protein or protein complexes were prepared in 5 µl volume, and deuterium exchange was initiated by addition of 45 µl of deuterated buffer (10 mM Tris-HCl pH 7.4/150 mM NaCl/10 mM DTT). For the condition of Clr4 with an excess of H3 peptide, the deuterated buffer was supplemented with H3 peptide (from lyophilized powder). Exchange was carried out at RT for four timepoints (3 s, 30 s, 300 s, 3000 s) and terminated by the addition of 20 µl ice-cold quench buffer (3 M guanidine – HCl/100 mM $NaH_2PO_4$ pH 2.5/1% formic acid). All experiments were repeated in duplicates. Samples were immediately frozen in liquid nitrogen and stored at –80 °C for up to 2 weeks.

Protein samples were thawed and injected in a UPLC system immersed in ice. The protein was digested via two immobilized pepsin columns (Thermo #23131), and peptides were collected onto a VanGuard precolumn trap (Waters). The trap was subsequently eluted and peptides separated with a C18, 300 Å, 1.7 µm particle size Fortis Bio column 100 × 2.1 mm over a gradient of 8–30%B over 18 min at 90 µl/min (Buffer A: 0.1% formic acid; buffer B: 100% acetonitrile/0.1% formic acid). Mass spectra were acquired on an Orbitrap Velos Pro (Thermo), for ions from 400 to 2200 m/z using an electrospray ionization source operated at 160 °C, 5 kV of ion spray voltage. Peptides were identified

by data-dependent acquisition after MS/MS, and data were analyzed by Mascot. Deuterium incorporation levels were quantified using HD examiner software (Sierra Analytics), and the quality of every peptide was checked manually. Results are presented as a percentage of maximal deuteration, with a theoretical maximal deuteration level of 86.3% . All experimental details and data of percentage deuterium incorporation for all peptides can be found in *Supplementary files 2–4*. Differences in exchange in a peptide were considered significant if they met all three of the following criteria: (1) for Clr4: ≥7 % change in exchange, ≥ 0.6 Da difference in exchange, and a p-value<0.05 using a two-tailed Student's t-test; and (2) for ubiquitin: ≥9% change in exchange, ≥1.3 Da difference in exchange, and a p-value<0.05 using a two-tailed Student's t-test.

## Isothermal titration calorimetry

ITC experiments were performed at 23 °C using a MicroCal VP-ITC calorimeter (Malvern Panalytical). All proteins and peptides used were dialyzed overnight against ITC buffer (25 mM sodium phosphate pH 8.0, 100 mM NaCl, 10% v/v glycerol, 0.5 mM TCEP) prior to experiments. 10 µl of the indicated Clr4 constructs at 50 µM were injected in 180 s time intervals in the cell containing the H3K14ub peptide at 5 µM. After subtracting heat enthalpies for titrations of the respective proteins into buffer, the ITC data were analyzed using the Origin software provided by the manufacturer.

## Molecular docking

Molecular docking was performed using Clr4KMT[192-490] (PDBID:1MVH), ubiquitin (PDBID:1UBQ), and a peptide comprising residues 7–17 of histone H3. The input peptide was derived by extension of the H3 peptide bound to *N. crassa* DIM-5 (PDBID:1PEG). Docking was performed with a local version of HADDOCK 2.0 using the interaction interface identified by HDX-MS as restraint.

## Crystallization of Clr4-3FA

Clr4-3FA was purified as described above. Clr4-3FA at 20 mg/ml was mixed with H3K14ub peptide and S-adenosyl-homocysteine to a ratio of 1:1.2:10 (protein:peptide:SAH) and diluted to a concentration of 10 mg/ml. Protein was diluted 1:1 with reservoir solution (71.4 mM MES, 28.6 mM imidazole, 20% PEG 10 K, 30 mM magnesium acetate, 6.6% v/v MPD, 6.6% v/v PEG 1000, 6.6% v/v PEG 3350) and incubated at 18 °C in sitting-drop vapor diffusion setup. Crystals were directly frozen in liquid nitrogen.

## Crystallographic data collection and structure determination

X-ray diffraction data were collected at the Diamond Light Source beamline I04 (Didcot, Oxford, UK) at a wavelength of 0.97950 Å. The datasets were analyzed using XDS and the CCP4 suite (*Kabsch, 2010*; *Winn et al., 2011*). Data were corrected for anisotropy using the STARANISO server (http://staraniso.globalphasing.org). Molecular replacement was performed using MOLREP and the coordinates from PDBID 6BOX (chain A) as search model (*Vagin and Teplyakov, 2010*). Refinement was performed using REFMAC, and Coot was used to build the Clr4-3FA model (*Emsley et al., 2010*; *Murshudov et al., 2011*). Structural alignments and figures were generated in PyMOL Molecular Graphics System, version 1.8, Schrödinger, LLC. Data collection and refinement statistics are presented in *Table 2*.

## Accession codes

Coordinates and structure factors have been deposited in the Protein Data Bank under accession code 6Z2A.

## Acknowledgements

We thank Janet Partridge, Robin Allshire, and Shiv Grewal for providing strains. We thank Steven Jacobsen, Chris Lima and Cheryl Arrowsmith for plasmids. We thank John Challiss, Jaswir Basran, and Raj Patel for support with enzymatic assays. We would like to thank Diamond Light Source for beamtime (proposal mx19880), and the staff of beamlines I04 for assistance with crystal testing and data collection. BF was supported by EPFL. OV was supported by a Swiss National Science Foundation (SNSF) Ambizione Fellowship (PZ00P3_148269). TS was supported by a SNSF Professorship (PP00P3_139137, PP00P3_163760_1, PP00P3_172904), a Biotechnology and Biological Sciences

Research Council Project Grant (BB/S018549/1) and contributions by Fondation Ernst et Lucie Schmidheiny, Fonds Constantin Topali and Société Académique de Genève.

## Additional information

### Funding

| Funder | Grant reference number | Author |
|---|---|---|
| Schweizerischer Nationalfonds zur Förderung der Wissenschaftlichen Forschung | PP00P3_139137 | Thomas Schalch |
| Schweizerischer Nationalfonds zur Förderung der Wissenschaftlichen Forschung | PP00P3_163760_1 | Thomas Schalch |
| Schweizerischer Nationalfonds zur Förderung der Wissenschaftlichen Forschung | PP00P3_172904 | Thomas Schalch |
| Schweizerischer Nationalfonds zur Förderung der Wissenschaftlichen Forschung | PZ00P3_148269 | Oscar Vadas |
| Biotechnology and Biological Sciences Research Council | BB/S018549 | Thomas Schalch |

The funders had no role in study design, data collection and interpretation, or the decision to submit the work for publication.

### Author contributions

Alessandro Stirpe, Data curation, Formal analysis, Investigation, Methodology, Validation, Writing – original draft, Writing – review and editing; Nora Guidotti, Sinan Kilic, Investigation, Methodology; Sarah J Northall, Data curation, Formal analysis, Investigation, Methodology, Validation; Alexandre Hainard, Resources; Oscar Vadas, Conceptualization, Data curation, Formal analysis, Investigation, Methodology, Resources, Validation, Visualization, Writing – original draft, Writing – review and editing; Beat Fierz, Conceptualization, Funding acquisition, Methodology, Project administration, Resources, Supervision, Validation, Writing – original draft, Writing – review and editing; Thomas Schalch, Conceptualization, Data curation, Formal analysis, Funding acquisition, Investigation, Methodology, Project administration, Resources, Supervision, Validation, Visualization, Writing – original draft, Writing – review and editing

### Author ORCIDs

Alessandro Stirpe (iD) http://orcid.org/0000-0002-2006-4066
Oscar Vadas (iD) http://orcid.org/0000-0003-3511-6479
Thomas Schalch (iD) http://orcid.org/0000-0002-0758-3013

### Decision letter and Author response

Decision letter https://doi.org/10.7554/eLife.62682.sa1
Author response https://doi.org/10.7554/eLife.62682.sa2

## Additional files

### Supplementary files

• Supplementary file 1. Oligonucleotide and peptide sequences.
 (A) Oligonucleotides used in qPCR experiments in *Figure 4*. (B) Peptide sequences used for synthesis of ubiquitinated H3 substrates. (C) sgRNA sequences used for CRISPR mutagenesis.

• Supplementary file 2. Hydrogen deuterium exchange mass spectrometry (HDX-MS) data for Clr4 alone and in complex with H3 or H3K14ub peptides.

• Supplementary file 3. Hydrogen deuterium exchange mass spectrometry (HDX-MS) data for H3K14ub alone and in complex with Clr4.

• Supplementary file 4. Hydrogen deuterium exchange mass spectrometry (HDX-MS) data collection statistics.

• Transparent reporting form

### Data availability

Coordinates and structure factors have been deposited in the Protein Data Bank under accession code 6Z2A.

The following dataset was generated:

| Author(s) | Year | Dataset title | Dataset URL | Database and Identifier |
|---|---|---|---|---|
| Stirpe A, Schalch T | 2021 | Structure of Clr4 mutant - F256A/F310A/F427A bound to SAH | https://www.rcsb.org/structure/6Z2A | RCSB Protein Data Bank, 6Z2A |

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
