## [Decision Letter]

**Acceptance summary:**

Using a combination of biochemistry and yeast genetics, Stirpe et al. show that the activity of the histone methyltransferase Clr4 (the fission yeast homolog of SUV39-family enzymes) is facilitated by interactions with another form of histone modification, ubiquitination of histone H3. Using hydrogen-deuterium exchange and structural modeling the authors identify the region of Clr4 that contacts the ubiquitin moiety on the H3 tail and design mutants to interfere with this function. The result is a Clr4 protein incapable of forming heterochromatic domains. Structural and biochemical analyses of homologs of Clr4 from other species suggested that this crosstalk might have been conserved during evolution. These findings shed new light on the intricate relationship between different histone marks and their roles in gene and genome regulation.

**Decision letter after peer review:**

Thank you for submitting your article "SUV39 SET domains mediate crosstalk of heterochromatic histone marks" for consideration by *eLife*. Your article has been reviewed by 3 peer reviewers, one of whom is a member of our Board of Reviewing Editors, and the evaluation has been overseen by Kevin Struhl as the Senior Editor. The following individual involved in review of your submission has agreed to reveal their identity: Jun-ichi Nakayama (Reviewer #2).

Based on the reviews and following discussion, the editors have judged your manuscript of interest but think that additional experiments are required before it is published.

Essential revisions

The following must be addressed in your revision:

1) Additional support for the claim that the mutants are only (or mostly) impaired in the ubiquitin binding activity. This is key for the proper interpretation of the in vivo data. As suggested by the reviewers, this could entail (but is not limited to) a better quantification or presentation of enzymatic activity (absolute instead of fold-change in stimulation), additional characterization of interacting proteins by mass spec, localization of the mutants to chromatin in a wild-type context.

2) Clarification of allostery vs. changes in binding affinities (Rev 1, point 4) ideally including measurements for the binding affinity of WT and mutants to the H3 peptide with and without ubiquitin.

3) Better characterization of silencing defects: ChIP-qPCR data should be included for both the dg and dh regions across mutants (Rev 3, point 4).

4) Analysis of the conservation of structural features in SUV34H2 (Rev 3 point 5).

Additional Points

We also think that several of the other points made by the reviewers might help you strengthen this manuscript and encourage you to consider addressing them if possible. The full reviews are included below.

*Reviewer #1:*

H3K14ub is a histone modification that facilitates deposition of H3K9me on heterochromatin in fission yeast, but the mechanism by which this modification stimulates Clr4 was unknown. Using mutants and HDX, the authors identified the interaction surface of Clr4 for H3K14ub, which they used to design mutants that responded poorly to H3K14ub stimulation. in vivo, these mutations resulted in loss of heterochromatin marks and defects in heterochromatin-based silencing, suggesting that H3K14ub stimulation is essential to K9me-mediated silencing. Finally, the authors show that human SUV39H2 but not G9a or Arabidopsis SUVH4 can be stimulated by H3K14ub in a similar manner.

The authors provided biochemical and structural insights into the mechanism that increases the H3K9-specific methyltransferase activity of Clr4 by H3K14ub. Although H3K14ub-mediated promotion of H3K9 methylation is shown in Oya et al., EMBO Rep 2019, this study further characterizes the potential mechanism. However, there are some issues with the results that need to resolved before I can recommend this for publication in *eLife*.

1. Similarity and difference with the previous study. As the authors acknowledge, this manuscript builds on a previous study by Oya et al., however I think the similarities and the differences need to be made even more explicit and better addressed.

1.1. The authors should clearly state that Figure 1B and 1C are basically a confirmation of Oya et al., 2019.

1.2. I am more puzzled by the difference in the mapping of the region required for H3K14ub stimulation. The authors suggest that a difference in the preparation of the recombinant proteins might be responsible. This can and should be tested as it would seemingly be a simple experiment (compare with and without GST tag).

1.3. Possibly to reconcile their findings with the previous report the authors state in the description of Figure 1 that "the N-terminus plays a regulatory role in the sensing of H3K14ub by the catalytic domain" but I don't see this reflected in the data show in Figure 1C, given that the degree of stimulation is very similar for KMT and FL.

2. Stimulation-defective mutants. The authors should carefully discuss the stimulation-defective mutants, which should be premised on the retention of their methyltransferase activity on unmodified H3. The authors claim that 30% loss of activity of the Clr4 KMT mutants on unmodified H3 is observed in Figure S3C (Pg 11 line 15), but this cannot be determined from the graph provided, which is normalized to unmodified H3. The authors should (1) make another graph to show the 30% loss and (2) compare Clr4 KMT mutants with catalytic-dead Clr4 KMT or dissolution buffer (no protein). It is still possible that GS253 and F3A mutations simply reduce MTase activity, thus displaying lower activity than WT in the presence of H3K14ub, which would also suggest a different interpretation for the results in vivo.

3. Heterochromatin localization of Clr4 mutants. The FLAG ChIP results in Figure 4E is not very informative, as with the loss of heterochromatin a loss of Clr4 is predicted. If the authors want to test whether the localization activity of Clr4 mutants is intact, (1) FLAG ChIP in the clr4+, Flag-Clr4GS253/F3A background (i.e., two clr4 alleles exist) or (2) in vitro H3K9me2/3 binding assay should be performed. Since Clr4 N-terminus might regulates MTase activity as discussed in Pg 18 line 19, it is also possible that amino acid substitutions in the KMT region affect the function of N-terminus, including CD. The co-IP in Figure 4C is not sufficient to clarify this point as Clr4 directly binds heterochromatin via its CD, in addition to the CLRC-mediated mechanism, and it is unclear if this is affected in the mutants.

4. Allosteric vs. binding regulation. On Pg. 11, the authors suggest that an allosteric mechanism is at play, but this is not supported by the data. In fact the observation that providing ubiquitin in trans does not stimulate and rather inhibits the activity on H3K14ub would suggest that the ubiquitin just increases binding affinity. To clarify this the authors should measure binding affinity of WT and mutants to the H3 peptide with and without ubiquitin.*Reviewer #2:*

In this manuscript Stirpe and colleagues describe structural insight into a novel regulation mechanism of SUV39 class histone methyltransferases. Clr4 is the sole SUV39-family H3K9me2/3 methyltransferase in fission yeast and recent evidence suggests that ubiquitylation of lysine 14 on histone H3 (H3K14ub) plays a key role in H3K9 methylation. To understand the molecular mechanisms of this regulation, the authors first set up in vitro assay system and demonstrate that H3K14ub promotes Clr4 methyltransferase activity and that the catalytic domain of Clr4 senses the presence of H3K14-linked ubiquitin. The authors then performed hydrogen/deuterium exchange coupled to mass spectrometry analysis and show that ubiquitin moiety binds to a region involving residues 243-261 of Clr4. Using this information, they further show that Clr4 mutants containing amino-acid substitutions in the ubiquitin binding region lose affinity for H3K14ub. The authors also demonstrate that fission yeast strains expressing mutant Clr4 display silencing defect and lose heterochromatic H3K9me2/3. Finally, the authors demonstrate that H3K14ub also stimulates the enzymatic activity of mammalian SUV39H2.

This is an excellent paper that provides structural insights into how H3K14ub stimulates Clr4 methyltransferase activity. The results presented are of high quality and convincingly controlled. The paper is carefully written, and the conclusions presented are fully supported by the data included. The results described are of high interest to the field of heterochromatin and crosstalk of histone marks. I recommend publication of this paper in *eLife*. The following points shall be addressed by the authors before publication.

Is the H3K14ub-mediated stimulation a shared property of SUV39 class methyltransferases? This is a quite important question considering the mechanisms underlying heterochromatin assembly in eukaryotic cells. While the authors demonstrate that SUV39H2's enzymatic activity is stimulated by H3K14u (Figure 5A), it would be interesting to test whether the activity of SUV39H1, the other mammalian Su(var)3-9 homologue, is also stimulated by the presence of H3K14ub.*Reviewer #3:*

The Suv39 class of methyl transferases are responsible for establishment and maintenance of constitutive heterochromatin via the deposition of H3K9me2/me3 marks. Clr4 is the sole H3K9me2/me3 HMTase in the fission yeast *S. pombe* and is part of the E3 ubiquitin ligase CLRC complex. It has been shown recently that CLRC mediates the ubiquitylation of H3K14 residue which in turn boosts the methyl transferase activity of Clr4. A region C-terminal to the chromo domain (aa 63-127) was also shown to be required to bind Ubiquitin and provide specificity for ubiquitylated H3K14 relative to inmodified H3 (Oya et al., 2019 EMBO Rep. 2019 20:e48111).

Here the authors further explore crosstalk between Clr4 activity and H3K14Ub. They do this via a structure-function approach employing a range of structural methods combined with in vivo assays. The primary finding here is that the presence of H3K14ub on histone H3 enhances Clr4 methyl transferase activity and this H3K14ub sensing region resides within the KMT methyltransferase domain itself (aa 192-490) not the aa 63-127 region as previously reported.

The authors further identify regions within this domain that are responsible for H3K14ub binding and Clr4 mutants which abrogate this interaction. These Clr4 mutants display dramatically reduced activity towards ubiquitylated peptide substrates. in vivo tests show that the same mutants exhibit silencing defects associated with almost a complete loss of H3K9me2/me3 from centromeric heterochromatin. Additionally, the authors show that H3K14ub sensing also appears to operate within the KMT domain of human SUV39H2 but not human G9a or Arabidopsis SUVH4.

Thus the key differences here from the Oya et al., 2019 study are the structural approaches employed and that Ubiquitin is sensed by the KMT methyltransferase domain itself without the previously identified Ubiquitin binding region in (aa 63-127). The authors offer a reasonable explanation for this discrepancy.

Additional analyses would perhaps help to strengthen their conclusions.

1. The relevance of the proposed mechanism in a cellular chromatin context is unclear. A significant fraction of H3K9me2/3 nucleosomes isolated from cells should also carry H3K14ub in cis. How frequently do K9Me2/3 and K14ub co-occur on nucleosomes in heterochromatin regions? This could explored by westerns with anti-H3K9me2 and or me3 – a mobility shift equivalent to monoubiquitylation should be visible.

2. The authors should consider including mutant peptide controls such as H3K9RK14ub to make sure what is detected here is indeed H3K9 methylation. Additionally, a completely unrelated substrate such as a ubiquitylated H4 N-terminal peptide could be used in the methyltransferase assays to strengthen the authors claims of specificity.

3. The IP-western (Figure 4C) shows association of Clr4 proteins with the Rik1 and suggesting that they are incorporated into the CLRC complex. However, a more rigorous test would be analysis these IPs by mass spectrometry to determine if the Clr4 GS253 and F3A mutant proteins are indeed assembled into a CLRC complex containing the other components.

4. The Clr4-F3A mutant appears to have a differential effect on the level of transcript generation from the dg and dh regions of centromeric repeats. For completeness ChIP-qPCR data should be included for both the dg and dh regions (currently only dh is assayed Figure 4 E) to determine if a difference is also detected.

5. Are similar structural features found in the SUV39H2 KMT domain to those shown for Clr4 (Figure 5C) that would also allow ubiquitin to dock? Does computational comparison between Suv39H2, Clr4, G9a and SUVH4 provide insight into similarities/differences?

---

## [Author Response]

Essential revisionsThe following must be addressed in your revision:1) Additional support for the claim that the mutants are only (or mostly) impaired in the ubiquitin binding activity.

We have performed thorough enzyme kinetics for wild type vs mutants, which are shown in Figure 3G and fitted parameters are tabulated in Table 1. These experiments establish that the mutants are severely affected in substrate binding specifically for the H3K14ub substrate.

Unfortunately, we ran into quality issues with the Cisbio kit, and we had to switch to Promega's MTase-glow Methyltransferase kit. While there are quantitative differences between the results obtained with the two kits they agree very well qualitatively.

2) Clarification of allostery vs. changes in binding affinities (Rev 1, point 4) ideally including measurements for the binding affinity of WT and mutants to the H3 peptide with and without ubiquitin.

We have clarified our interpretation of H3K14ub's effect on Clr4 with the following changes to the text on p. 5. This interpretation is well supported by the comprehensive kinetic analysis discussed in (1). Measuring affinities of the unmodified peptide directly by ITC failed because the Kds are very high and require concentrations that we could not reach. Change to manuscript:

"Comparison of the kinetic parameters between H3K14ub and H3 substrate indicates that the presence of ubiquitin on lysine 14 leads to a tighter enzyme-substrate complex and to conformational changes in the active site that increase the rate of the methyltransferase reaction.

To determine whether H3K14ub uses an allosteric site for ubiquitin on Clr4, we challenged the methyltransferase reaction with increasing amounts of free ubiquitin. While we observed no significant increase in activity for unmodified H3, we observed a drop in the activity for H3K14ub at high concentrations of free ubiquitin (Figure 1E). This experiment failed to produce evidence of an allosteric site for free ubiquitin on Clr4 and we conclude that the stimulation of k_cat_ is likely to depend on an induced-fit mechanism triggered by binding of H3K14ub to the Clr4 HMT domain."

3) Better characterization of silencing defects: ChIP-qPCR data should be included for both the dg and dh regions across mutants (Rev 3, point 4).

We have further characterized the UBR mutants with ChIP-qPCR data for dg, dh and tlh1, and have added them to Figure 4E. These results do not show a differential effect in dg vs. dh for the Clr4FA mutant.

4) Analysis of the conservation of structural features in SUV34H2 (Rev 3 point 5)

We have added a comprehensive sequence and motif analysis in Figure S5D and E with the following text added to the end of the discussion:

"Bioinformatic analysis of the UBR sequence using Hidden Markov Models suggests that the UBR sequence is well conserved in the *Ascomycetes* clade of fungi, which includes the *N. crassa* Dim-5 protein for example (Figure S5D, E). Comparing the Clr4 motif with motifs obtained using homologous sequences from human SUV39H2, human G9a and *Arabidopsis* SUVH4 shows that SUV39H2's motif is very similar to Clr4, while G9a and SUVH4 diverge significantly. This is consistent with our observation that H3K14ub can stimulateSUV39H2, but not G9A or SUVH4."

Reviewer #1:1. Similarity and difference with the previous study. As the authors acknowledge, this manuscript builds on a previous study by Oya et al., however I think the similarities and the differences need to be made even more explicit and better addressed.1.1. The authors should clearly state that Figure 1B and 1C are basically a confirmation of Oya et al., 2019.

We have added: "These experiments confirm the observation by Oya et al., ^2^ that the H3K14ub substrate triggers a dramatic and specific increase in the methyltransferase activity of Clr4. However, in contrast to the previous study, we observe that the KMT domain is sufficient to mediate this regulatory mechanism."

1.2. I am more puzzled by the difference in the mapping of the region required for H3K14ub stimulation. The authors suggest that a difference in the preparation of the recombinant proteins might be responsible. This can and should be tested as it would seemingly be a simple experiment (compare with and without GST tag).

We agree that we cannot explain the discrepancy satisfactorily. However, Shan et al. ^1^ have completely independently confirmed our result and we therefore chose to focus our resources on characterizing the mutants.

1.3. Possibly to reconcile their findings with the previous report the authors state in the description of Figure 1 that "the N-terminus plays a regulatory role in the sensing of H3K14ub by the catalytic domain" but I don't see this reflected in the data show in Figure 1C, given that the degree of stimulation is very similar for KMT and FL.

We agree that our data do not establish the statistical significance to make this claim firmly and have therefore withdrawn the sentence.

2. Stimulation-defective mutants. The authors should carefully discuss the stimulation-defective mutants, which should be premised on the retention of their methyltransferase activity on unmodified H3.

We have addressed this by the kinetic analysis of the mutants in Figure 3G.

3. Heterochromatin localization of Clr4 mutants. The FLAG ChIP results in Figure 4E is not very informative, as with the loss of heterochromatin a loss of Clr4 is predicted. If the authors want to test whether the localization activity of Clr4 mutants is intact, (1) FLAG ChIP in the clr4+, Flag-Clr4GS253/F3A background (i.e., two clr4 alleles exist) or (2) in vitro H3K9me2/3 binding assay should be performed. Since Clr4 N-terminus might regulates MTase activity as discussed in Pg 18 line 19, it is also possible that amino acid substitutions in the KMT region affect the function of N-terminus, including CD. The co-IP in Figure 4C is not sufficient to clarify this point as Clr4 directly binds heterochromatin via its CD, in addition to the CLRC-mediated mechanism, and it is unclear if this is affected in the mutants.

We agree that the FLAG ChIP exclusively reports on the presence of Clr4 in heterochromatic regions and that the data confirm what is expected when heterochromatin is lost. We also agree that the proposed experiments would be very interesting and could potentially provide revealing insight into the recruitment of Clr4. However, we are of the opinion that dissecting the contribution of CD, H3K14ub and potentially other mechanisms to Clr4 recruitment goes beyond the scope of this manuscript, which is focused on the enzymatic stimulation of Clr4 through H3K14ub.

4. Allosteric vs. binding regulation. On Pg. 11, the authors suggest that an allosteric mechanism is at play, but this is not supported by the data. In fact the observation that providing ubiquitin in trans does not stimulate and rather inhibits the activity on H3K14ub would suggest that the ubiquitin just increases binding affinity. To clarify this the authors should measure binding affinity of WT and mutants to the H3 peptide with and without ubiquitin.

This point has been addressed in Essential Points 2.

Reviewer #2:Is the H3K14ub-mediated stimulation a shared property of SUV39 class methyltransferases? This is a quite important question considering the mechanisms underlying heterochromatin assembly in eukaryotic cells. While the authors demonstrate that SUV39H2's enzymatic activity is stimulated by H3K14u (Figure 5A), it would be interesting to test whether the activity of SUV39H1, the other mammalian Su(var)3-9 homologue, is also stimulated by the presence of H3K14ub.

We agree that it would be very interesting to screen the SUV39 enzyme family to determine which members share the H3K14ub-mediated stimulation. However, we feel that addressing this question is beyond the scope of this manuscript.

Reviewer #3:1. The relevance of the proposed mechanism in a cellular chromatin context is unclear. A significant fraction of H3K9me2/3 nucleosomes isolated from cells should also carry H3K14ub in cis. How frequently do K9Me2/3 and K14ub co-occur on nucleosomes in heterochromatin regions? This could explored by westerns with anti-H3K9me2 and or me3 – a mobility shift equivalent to monoubiquitylation should be visible.

Oya et al., have addressed this question by showing that H3K14ub can be detected in pulldown of H3K9me2. Doing this quantitatively is extremely difficult since H3K14ub is very likely removed efficiently by deubiquitinases during isolation procedures. We would like to point to Shan et al., ^1^ who provide further genetic evidence that the H3K14ub modification is critical for H3K9me2/3 in a physiological context.

2. The authors should consider including mutant peptide controls such as H3K9RK14ub to make sure what is detected here is indeed H3K9 methylation. Additionally, a completely unrelated substrate such as a ubiquitylated H4 N-terminal peptide could be used in the methyltransferase assays to strengthen the authors claims of specificity.

We agree that our manuscript does not fully address the question of specificity and have adjusted the wording accordingly. (see response to Rev 1, point 3.)

3. The IP-western (Figure 4C) shows association of Clr4 proteins with the Rik1 and suggesting that they are incorporated into the CLRC complex. However, a more rigorous test would be analysis these IPs by mass spectrometry to determine if the Clr4 GS253 and F3A mutant proteins are indeed assembled into a CLRC complex containing the other components.

The IP-Westerns clearly show that Clr4 remains associated with CLRC even though heterochromatin is lost. We believe that hunting after potential secondary consequences for the CLRC complex is beyond the scope of this manuscript.

4. The Clr4-F3A mutant appears to have a differential effect on the level of transcript generation from the dg and dh regions of centromeric repeats. For completeness ChIP-qPCR data should be included for both the dg and dh regions (currently only dh is assayed Figure 4 E) to determine if a difference is also detected.

We have included the requested ChIP-qPCR data in Figure 4E.

5. Are similar structural features found in the SUV39H2 KMT domain to those shown for Clr4 (Figure 5C) that would also allow ubiquitin to dock? Does computational comparison between Suv39H2, Clr4, G9a and SUVH4 provide insight into similarities/differences?

We have included a corresponding sequence analysis in Figure S5D, E.